# Combined comparative genomics and clinical modeling reveals plasmid-encoded genes are independently associated with *Klebsiella* infection

Jay Vornhagen[1,2], Emily K. Roberts [3], Lavinia Unverdorben[2], Sophia Mason[1], Alieysa Patel[1], Ryan Crawford[4], Caitlyn L. Holmes[1,2], Yuang Sun [1], Alexandra Teodorescu[1], Evan S. Snitkin[2,5], Lili Zhao[3], Patricia J. Simner[6], Pranita D. Tamma[7], Krishna Rao[5], Keith S. Kaye[5] & Michael A. Bachman [1,2] ✉

Members of the *Klebsiella pneumoniae* species complex frequently colonize the gut and colonization is associated with subsequent infection. To identify genes associated with progression from colonization to infection, we undertook a case-control comparative genomics study. Concordant cases ($N = 85$), where colonizing and invasive isolates were identical strain types, were matched to asymptomatically colonizing controls ($N = 160$). Thirty-seven genes are associated with infection, 27 of which remain significant following adjustment for patient variables and bacterial phylogeny. Infection-associated genes are not previously characterized virulence factors, but instead a diverse group of stress resistance, regulatory and antibiotic resistance genes, despite careful adjustment for antibiotic exposure. Many genes are plasmid borne, and for some, the relationship with infection is mediated by gut dominance. Five genes were validated in a geographically-independent cohort of colonized patients. This study identifies several genes reproducibly associated with progression to infection in patients colonized by diverse *Klebsiella*.

The *Klebsiella pneumoniae* species complex (hereby referred to as *Klebsiella*) is comprised of several species capable of causing severe infections, including bacteremia, pneumonia, and urinary tract infection (UTI). The member species are *K. pneumoniae*, *K. variicola*, *K. quasipneumoniae*, *K. quasivariicola* sp. nov.[1], and *K. africana*[2], which are genetically distinct but often clinically indistinguishable (reviewed in[3]). *Klebsiella* infections are a serious public health concern because they are a leading cause of healthcare-associated infections[4], can

harbor multiple antimicrobial resistance (AMR) determinants, and can be hypervirulent. Antimicrobial resistant *Klebsiella* complicate disease treatment, leading to high mortality and healthcare costs whereas hypervirulent *Klebsiella* cause unique clinical manifestations such as pyogenic liver abscess and meningitis[3,5,6]. *Klebsiella* colonization is the primary risk factor for disease[7,8], where the gut acts as a reservoir for disease-causing *Klebsiella* strains[7,9]. Patient variables, including comorbidities and baseline laboratory values, partially explain which

[1]Department of Pathology, Michigan Medicine, University of Michigan, Ann Arbor, MI, USA. [2]Department of Microbiology & Immunology, Michigan Medicine, University of Michigan, Ann Arbor, MI, USA. [3]Department of Biostatistics, School of Public Health, University of Michigan, Ann Arbor, MI, USA. [4]Department of Computational Medicine and Bioinformatics, Michigan Medicine, University of Michigan, Ann Arbor, MI, USA. [5]Department of Internal Medicine/Infectious Diseases Division, Michigan Medicine, University of Michigan, Ann Arbor, MI, USA. [6]Division of Medical Microbiology, Department of Pathology, Johns Hopkins University School of Medicine, Baltimore, MI, USA. [7]Department of Pediatrics, Johns Hopkins University School of Medicine, Baltimore, MI, USA. ✉e-mail: mikebach@med.umich.edu

patients will progress to infection[10]. Furthermore, the high relative abundance of *Klebsiella* colonizing the gut is associated with an increased risk of infection in colonized patients[11]. For hypervirulent *Klebsiella*, the genes associated with their unique presentations have been well-defined by molecular epidemiology and experimental studies (reviewed in[12]); however, little is known about the *Klebsiella* genes contained by non-hypervirulent strains, including multi-drug resistant (MDR) strains, that increase the risk of disease in colonized patients more broadly. Given that most infections are caused by non-hypervirulent strains[13], more data concerning the genetic determinants of infection are necessary for predicting, diagnosing, and experimentally evaluating these strains.

In general, the ability of bacteria to progress from colonization to infection is dependent on the immune status of the host, environmental and iatrogenic exposures such as antimicrobials, and the virulence potential of the strain. Known *Klebsiella* virulence determinants are found in both the core genome, which are -1700 genes present in >95% of all sequenced strains, and the accessory genome, which are genes that vary among sequenced strains[14]. The overall *Klebsiella* accessory genome is currently estimated to include >100,000 unique genes[3]. With an average of >5000 genes per genome, thousands of genes vary between strains. The size of the *Klebsiella* accessory genome provides a unique opportunity to perform comparative genomics studies to understand pathogenesis. Comparative genomics approaches are used to understand hospital outbreaks[15,16], identify in-host adaptation[17], and infer conserved virulence determinants or pathways. The latter approach has been used to interrogate the pathogenicity of numerous bacterial pathogens, such as *Pseudomonas aeruginosa*[18], *Streptococcus pneumoniae*[19], *Escherichia coli*[20], and invasive non-typhoidal *Salmonella enterica* spp.[21]. Previously, we used a comparative genomics approach to identify *Klebsiella* factors associated with infection in a small, hospital-wide cohort of *Klebsiella*-colonized patients[22]. Two infection-associated factors identified in this study, the psicose utilization locus and *ter* operon, were experimentally validated and characterized[22,23], confirming the value of this approach for understanding *Klebsiella* pathogenesis.

The strong association between gut colonization and *Klebsiella* disease indicates a potential point of intervention to prevent infection in the most at-risk patients, such as those in intensive care and hematology/oncology units. If patients with risk factors for infection are colonized by high-risk *Klebsiella* isolates, early detection and intervention may prevent disease. The development of these approaches is likely to be critical as therapeutic options become limited due to rising antimicrobial resistance. We previously performed a large cohort study of *Klebsiella*-colonized patients in the intensive care and hematology/oncology units and identified patient variables associated with subsequent infection[10]. Additionally, we identified an association between gut dominance by *Klebsiella* and infection, even after adjustment for the patient variables[11]. Here, we aimed to rigorously identify *Klebsiella* genes that are associated with disease in colonized patients through whole-genome sequencing (WGS) in a nested case-control study from a cohort of over 1900 colonized patients. This approach differed from other comparative genomics studies in that it focused on concordant infections (e.g., same colonizing and infecting isolate) after infection, directly compared cases of infection to well-matched controls from the same patient population and incorporated careful adjustment with patient variables associated with infection. Through this approach, we identified 27 genes associated with infection in colonized patients. We identified which genes may act through gut dominance, and determined that many genes were found on large, conjugative plasmids. Finally, we demonstrate that five of these genes are associated with infection in a geographically-independent cohort of intensive care patients colonized with *Klebsiella*.

## Results
### Selection of *Klebsiella* for WGS and description of the case-control study
A nested case-control study was performed using a cohort of 1978 *Klebsiella* colonized, intensive care, and hematology/oncology patients at a single academic medical center in Michigan, USA[10]. Over the course of this study patient rectal swabs were screened for the presence of *Klebsiella*. If a patient was colonized with *Klebsiella*, up to three rectal isolates were banked, and the patient was enrolled in the study. Subsequent rectal swab isolates and clinical culture *Klebsiella* isolates (blood, respiratory, urinary, and other) were also banked (Fig. 1a). Cases of infection were considered for inclusion based on chart review using strict clinical definitions[10]. To facilitate a rigorous

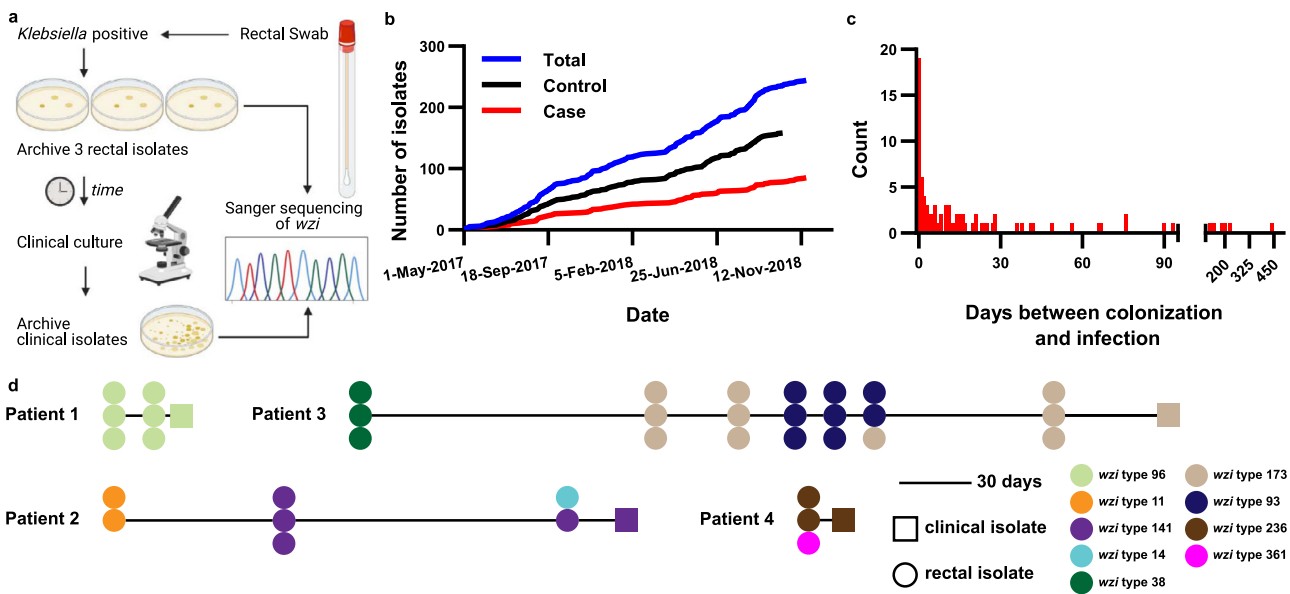

**Fig. 1 | Description of *Klebsiella* isolate collection. a** Schematic representation of *Klebsiella* isolate collection and *wzi*-based concordance testing for WGS selection. **b** Rectal isolate selected for WGS collection dates stratified by case status. **c** Histogram of the number of days between identification of a rectal *Klebsiella* isolate concordant with the clinical isolate from the same patient. **d** Trace diagrams of *wzi* typing in *Klebsiella* colonizing and infecting isolates in infected patients.

**Table 1 | Patient demographics and select baseline characteristics**

| Patient variable | | Case (N = 85) | Control (N = 160) | Odds ratio (OR)* | 95% CI | Fisher P-value |
|---|---|---|---|---|---|---|
| Sex | Female | 40 (47.1%) | 76 (47.5%) | | | |
| | Male | 45 (52.9%) | 81 (50.6%) | | | |
| | Missing | 0 (0.0%) | 3 (1.88%) | | | |
| Median age (range) | | 60 (83–25) | 59 (85–27) | | | |
| Infection site | Blood | 39 (45.9%) | | | | |
| | Respiratory | 19 (22.4%) | | | | |
| | Urine | 23 (27.1%) | | | | |
| | Other | 4 (4.71%) | | | | |
| Mean weighted Elixhauser score (±SD) | | 22.2 ± 11.6 | 19.7 ± 11.9 | 1.02 | 0.994 to 1.04 | 0.127 |
| Depression | Yes | 30 (35.3%) | 44 (27.5%) | 1.31 | 0.761 to 2.26 | 0.327 |
| | No | 55 (64.7%) | 113 (70.6%) | | | |
| | Missing | 0 (0.0%) | 3 (1.88%) | | | |
| Prior diuretics | Yes | 30 (35.3%) | 38 (23.8%) | 1.69 | 0.905 to 3.05 | 0.101 |
| | No | 55 (64.7%) | 119 (74.4%) | | | |
| | Missing | 0 (0.0%) | 3 (1.88%) | | | |
| Prior Vitamin D | Yes | 18 (21.2%) | 19 (11.8%) | 1.66 | 0.972 to 4.44 | 0.0591 |
| | No | 67 (78.8%) | 138 (86.3%) | | | |
| | Missing | 0 (0.0%) | 3 (1.88%) | | | |
| Prior pressors/inotrope | Yes | 20 (23.5%) | 14 (8.75%) | 3.06 | 1.40 to 6.68 | **0.00502** |
| | No | 65 (76.6%) | 143 (89.4%) | | | |
| | Missing | 0 (0.0%) | 3 (1.88%) | | | |
| High-risk antibiotics | Yes | 30 (35.3%) | 32 (20.0%) | 2.21 | 1.18 to 4.14 | 0.013 |
| | No | 55 (64.7%) | 125 (78.1%) | | | |
| | Missing | 0 (0.0%) | 3 (1.88%) | | | |
| Albumin | <2.5 g/dL | 35 (41.2%) | 34 (21.3%) | 2.86 | 1.49 to 5.48 | **0.00157** |
| | ≥2.5 g/dL | 47 (55.3%) | 116 (72.5%) | | | |
| | Missing | 3 (3.53%) | 10 (6.25%) | | | |

*Two-sided test. No adjustments were made for multiple comparisons. OR for sex was incalculable due to use of sex as a matching variable.
Bold text indicates significant results (P-value ≤ 0.01 after adjustment for clinical variables).

comparative genomics approach, Sanger sequencing of the *wzi* gene[24] was used to screen for concordance between the clinical culture isolate and a preceding colonizing rectal isolate from the same patient (Fig. 1a). Cases and controls accumulated at a similar rate throughout the enrollment period (Fig. 1b). The median time between detection of *Klebsiella* colonization and subsequent infection was 10 days (range = 0–443 days, Fig. 1c). In some instances, the progression from colonization to infection was simple, wherein a single *wzi* type was detected in all colonizing *Klebsiella* isolates and the subsequent infecting isolate (Fig. 1d, "Patient 1"). In other instances, the infection trajectory was more complicated, wherein some patients exhibited single or multiple strain replacements in the gut prior to infection (Fig. 1d, "Patient 2" and "Patient 3"), or mixed colonization where only one colonizing isolate caused the subsequent infection (Fig. 1d, "Patient 2" and "Patient 4"). Collectively, these data indicate concordant infections are detectable in colonized patients in the context of highly dynamic *Klebsiella* populations.

In total, 85 cases of infection with an identical *wzi* type to a colonizing rectal isolate (cases) were selected for WGS, including 39 bloodstream infections, 23 urinary tract infections, 19 pneumonias, and 4 other sites of infection (including two cases of cholecystitis, one case of empyema, and one case of ecthyma). Based on antibiotic susceptibility data from medical records, 44 of 85 (51.8%) of the clinical isolates from cases were pan-sensitive apart from intrinsic ampicillin resistance. The most common AMR phenotypes were resistance to first- and second-generation cephalosporins, the combinatorial β-lactam/β-lactamase-inhibitors ampicillin-sulbactam and piperacillin-

tazobactam, and trimethoprim-sulfamethoxazole (Supplementary Fig. S1). Cases were matched to two colonized controls (who remained asymptomatic) based on sex, age, and date of rectal swab collection. In addition, controls had the same type of clinical culture collected as their corresponding case, but with no cultivatable bacteria. Two strata included only a single control because of the inability to find an appropriate match, and an additional 8 controls were removed from the study following WGS based on species classification outside of *Klebsiella*. In the final case-control design, 85 cases, including the clinical and rectal isolates, and 160 control rectal isolates were evaluated (Table 1).

### Description of sequenced *Klebsiella* isolates
Following the determination of isolates for WGS, genomic DNA was extracted from all *Klebsiella* isolates and subjected to Illumina sequencing (Supplementary Data 1). Assembled genomes were generally of high quality, wherein the median number of contigs was 110.5 (range = 43–876 contigs) and the median N50 was 459,591 bp (range = 147,972–3,285,432 bp). All genomes were of expected length, wherein the median genome size was 5.50 Kb (range = 5.06–6.26 Kb). Sequencing quality was similar between rectal case and control isolates, which were used for most subsequent analyses (Supplementary Fig. S2). As expected, species assignment based on WGS included the three most common members of the *Klebsiella pneumoniae* species complex (*K. pneumoniae, K. variicola, K. quasipneumoniae*). *K. pneumoniae, K. variicola*, and *K. quasipneumoniae* were evenly distributed amongst the four sites of infection and control rectal swabs

(Supplementary Fig. S3a). Case and control rectal isolates within a stratum may be the same or different species, as reflected in either high (100–80%) or low (60–30%) overall nucleotide identity (Supplementary Fig. S3b). For the rectal and clinical isolate of a given case of infection, matching *wzi* types were highly predictive of overall nucleotide identity (median % nucleotide identity = 99.65%, range = 95.04–99.88%, Supplementary Fig. S3c). The time between detection of *Klebsiella* colonization and subsequent infection was negatively correlated with sequence similarity; however, the effect was minimal ($r^2$ = 0.06, Supplementary Fig. S3d). Interestingly, instances of lower nucleotide identity similarity (<98%) between the rectal and clinical isolates from concordant cases of infection appeared to be due to large genomic losses or additions (Supplementary Fig. S3e), which may be driven by plasmids or other mobile genetic elements (MGEs). Indeed, these case clinical and rectal isolate pairs displayed differences in the number of annotations that corresponded to differences in genome size (Supplementary Fig. S3f–g).

To confirm concordance between case clinical and rectal isolates, *wzi* strain typing was performed from the assembled genomes. All *wzi* types matched results from Sanger sequencing, and construction of a *wzi* approximately-maximum-likelihood tree revealed a total of 147 unique *wzi* types identified (Supplementary Fig. S4). Then, multi-locus sequence typing was performed across all isolates. One-hundred thirty-one unique sequence types (STs) were identified, and all case rectal and case clinical STs were identical. Rarefaction of STs indicated that cases and controls were equally rich (Supplementary Fig. S5). ST253 was the most abundant ST but still low frequency (10/245 rectal isolates, 4.08%), and 44 rectal isolates were non-typable (44/245, 17.96%). Overall, hypervirulent clonal groups (CGs)[25] were rare, comprising 2.35% of cases and 1.35% of controls. Only two isolates of hypervirulent ST23 were identified, and only one caused an infection. MDR CGs[25], frequently containing extended-spectrum β-lactamases, carbapenemases, or both, were more common, representing 18.8% of cases and 9.38% of controls. These included 1 ST11, 4 ST14, 3 ST15, 1 ST16, 3 ST17, 5 ST37, 2 ST147, 1 ST152, 1 ST258, and 3 ST307. 85.7% (210/245) of the isolates in our collection were from non-hypervirulent, non-MDR CGs. Combined, MDR CGs were significantly associated with case status (odds ratio = 2.24, 95% C.I. 1.03 to 4.93, Fisher $P$ = 0.043). Construction of a core genome phylogeny demonstrated the presence of three *Klebsiella* species within our cohort and multiple lineages across colonizing and invasive isolates of each species (Supplementary Data 1; Fig. 2a). Together, these data indicate that sequenced *Klebsiella* isolates were highly diverse, without a dominant hypervirulent or MDR CG within the population.

To enable the identification of infection-associated genes in rectal samples, we assembled a pangenome (Supplementary Data 2). The rectal isolate pangenome consisted of 26,089 total annotations. The core genome (genes present in ≥95% genomes) was comprised of 3921 genes, and the accessory genome (genes present in <95% genomes) consisted of 22,168 genes (Supplementary Fig. S6a). To assess the similarity of the accessory genome between *Klebsiella* isolates in our collection, pairwise Jaccard distances were calculated after excluding the core genome and summarized as a mean for each isolate (244 pairwise comparisons per isolate). Interestingly, the mean pairwise Jaccard distance for all rectal isolates was 0.73 ± 0.04, indicating that the accessory genome of each isolate was highly dissimilar from other isolates. No differences were observed between mean pairwise Jaccard distances for cases and controls (Supplementary Fig. S6b). Overall, the cohort of *Klebsiella* isolates exhibited high pangenome diversity.

## Genes associated with *Klebsiella* infection

To identify genes in the accessory genome associated with infection we used Scoary, collapsing linked genes that displayed identical presence/absence patterns across all sequenced isolates (e.g., organized in operons, linked to MGEs, etc.) into units and comparing unit frequencies between cases and controls (Supplementary Data 3)[26]. Thirty-seven genes across 34 units were significantly associated with infection (*Q*-value ≤ 0.2, 2166 tests). Two units contained genes with identical presence/absence patterns (*group_5875-tehA_2-group_10494* and *group_5455-sul2*); the remaining units contained a single gene. Overall, 36 genes were associated with cases, one was associated with controls (Table 2).

To control for patient variables in our comparative genomics analysis, we leveraged a previously constructed multivariate model for infection risk in the larger cohort from which this nested case-control study was derived. This model was comprised of mean weighted Elixhauser comorbidity score, depression, prior diuretic use, vitamin D use, pressors/inotrope use, high-risk antibiotic use, and low albumin. Prior medication use was defined as >48 h but <90 days prior to the colonizing swab and albumin was measured within 48 h of the colonizing swab[10]. High-risk antibiotics, based on their impact on indigenous gut microbiota, were defined as β-lactam/β-lactamase inhibitor combinations, carbapenems, third- and fourth-generation cephalosporins, fluoroquinolones, clindamycin, and oral vancomycin[27]. Consistent with the explanatory value of these variables in the overall cohort, the variables used for adjustment trended towards or were significantly associated with infection in the nested case-control study (Table 1). We then used inverse probability of treatment weighting to adjust for these variables in measuring the association of each gene with the outcome of infection.

Following adjustment for clinical variables including comorbidities and antibiotic exposure, 28 of 37 genes remained significantly associated with infection (*P*-value ≤ 0.01; Table 2, Fig. 2b). These 28 infection-associated genes fell into four categories. First, many infection-associated genes encode AMR proteins even after correction for exposure to high-risk antibiotics. These included *aac(6′)-Ib-cr5*, encoding an aminoglycoside acetyltransferase that also has activity against fluoroquinolones[28], and *bla*<sub>OXA-1</sub>, *bla*<sub>CTX-M-15</sub>, *bla*<sub>TEM-1</sub> encoding broad-spectrum β-lactamases. Second, several infection-associated genes have predicted protein functions that may alter the physiology of the bacteria directly. These include the predicted aquaporin gene *aqpZ_2*, tellurite-resistance gene *tehA_2*, arsenate resistance gene *arsD_1*, and diguanylate cyclase gene *dgcE*. Third, certain MGEs and plasmid maintenance genes were associated with infection (e.g., IS1380 family transposase ISEcp1, Transposon Tn3 resolvase, IS6 family transposase IS6100, *umuDC_2*). The remainder of infection-associated genes encode hypothetical proteins. Interestingly, certain genes were highly associated with one another, suggesting the presence of linked genes that may be acquired simultaneously through horizontal gene transfer (HGT) events (Supplementary Fig. S7).

None of the *Klebsiella* genes associated with infection encoded known virulence factors. To directly assess the association of known virulence genes with infection, we used the Kleborate genotyping tool, which is an in silico platform for rapid genotyping of *Klebsiella* genomes (Supplementary Data 4)[25]. Consistent with the paucity of hypervirulent STs, most well-characterized *Klebsiella* virulence factors were rare in our dataset, including salmochelin (*iro*), aerobactin (*iuc*), and colibactin (*clb*) synthesis loci, and hypermucoviscosity regulators RmpA/A2 (Table 3). The yersiniabactin (*ybt*) synthesis locus was more frequent overall, and in cases versus controls, but was not associated with infection (odds ratio = 1.7, *P* = 0.144). Correspondingly, the virulence score, which is a composite metric of the virulence loci listed above, was not significantly different between cases and controls (Case virulence score = 0.294 ± 0.784, Control virulence score = 0.163 ± 0.513, *P* = 0.115, Student's *t*-test). Even when assessed directly and after controlling for clinical variables, canonical virulence factors are not associated with infection in this group of colonized, intensive care patients.

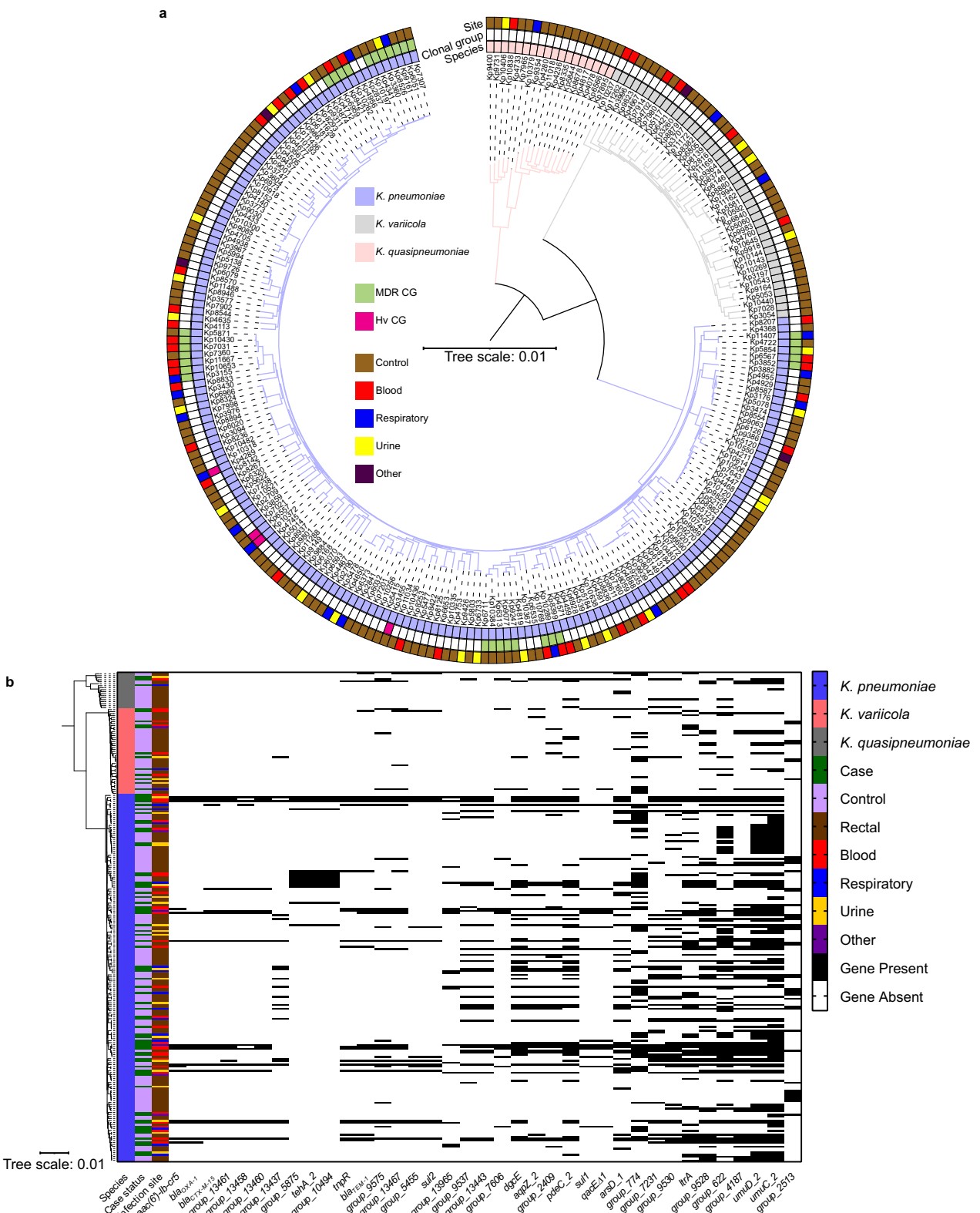

**Fig. 2 | *Klebsiella* rectal isolates are highly diverse and infection-associated genes span that diversity. a** Core genome phylogeny tree of sequenced *Klebsiella* rectal isolates. The inner ring indicates the species, the middle ring indicates a clonal group of concern (hypervirulent [Hv] CG: clonal groups 23, 25, 380; MDR CG: 14/15, 17/20, 37, 147, 258, 307), and the outer ring indicates site of infection for concordant clinical isolate. **b** Heatmap of infection-associated gene presence/ absence organized vertically by core genome phylogeny and horizontally by the strength of association (Odds ratio) between gene presence and infection following adjustment for patient variables.

**Table 2 | Genes associated with patient case status**

| Gene | Annotation | Case frequency | Control frequency | Unadjusted | | | Adjusted for clinical variables | | Mediation | | | |
|---|---|---|---|---|---|---|---|---|---|---|---|---|
| | | | | Odds ratio | P-value | Q-value** | Odds ratio | P-value** | Direct β | Direct P-value** | Indirect β | Indirect P-value** |
| aac(6')-Ib-cr5 | fluoroquinolone-acetylating aminoglycoside 6'-N-acetyltransferase AAC(6')-Ib-cr5 | 0.176 | 0.006 | 30.68 | **0.0011** | **0.00131** | 52.41 | **0.0003** | 3.0601 | **0.0045** | 0.086 | **0.01** |
| bla_OXA-1 | oxacillin-hydrolyzing class D β-lactamase OXA-1 | 0.153 | 0.006 | 25.54 | **0.002** | **0.0053** | 49.29 | **0.0003** | 2.9698 | **0.0064** | 0.077 | 0.03 |
| bla_CTX-M-15 | class A extended-spectrum β-lactamase CTX-M-15 | 0.165 | 0.013 | 13.94 | **0.0007** | **0.0053** | 19.17 | **0.0004** | 2.3386 | **0.004** | 0.047 | 0.1567 |
| group_13461 | IS1380 family transposase ISEcp1 | 0.165 | 0.013 | 13.94 | **0.0007** | **0.0053** | 18.69 | **0.0003** | 2.482 | **0.0022** | 0.019 | 0.5833 |
| group_13458 | hypothetical protein | 0.141 | 0.013 | 11.46 | **0.0018** | **0.01959** | 17.57 | **0.0006** | 2.0832 | 0.0112 | 0.034 | 0.3483 |
| group_13460 | hypothetical protein | 0.153 | 0.013 | 12.69 | **0.0011** | **0.01065** | 16.73 | **0.0005** | 2.3118 | **0.0047** | 0.024 | 0.5217 |
| group_13437 | hypothetical protein | 0.176 | 0.031 | 5.97 | **0.001** | **0.05505** | 7.28 | **0.0014** | 1.6137 | **0.0054** | 0.005 | 0.875 |
| group_5875-tehA_2-group_10494* | HTH-type transcriptional regulatory protein GabR - tellurite-resistance protein TehA - hypothetical protein | 0.129 | 0.025 | 5.65 | **0.004** | 0.1908 | 7.09 | **0.0029** | 1.8251 | **0.0053** | 0.007 | 0.8383 |
| tnpR | transposon Tn3 resolvase | 0.176 | 0.031 | 5.97 | **0.001** | **0.05505** | 6.24 | **0.0017** | 1.6379 | **0.0054** | 0.023 | 0.45 |
| bla_TEM-1 | class A broad-spectrum β-lactamase TEM-1 | 0.212 | 0.038 | 6.28 | **0.0002** | **0.01236** | 5.46 | **0.0009** | 1.6623 | **0.0021** | 0.035 | 0.1783 |
| group_9575 | tyrosine recombinase XerC | 0.259 | 0.056 | 5.39 | **0.0001** | **0.0053** | 4.72 | **0.0019** | 1.3993 | **0.0044** | 0.048 | 0.05 |
| group_13467 | IS6 family transposase IS6100 | 0.224 | 0.044 | 5.75 | **0.0002** | **0.01093** | 4.58 | **0.0026** | 1.5338 | **0.003** | 0.023 | 0.4017 |
| group_5455-sul2* | IS91 family transposase - sulfonamide-resistant dihydropteroate synthase Sul2 | 0.129 | 0.025 | 5.07 | **0.0077** | 0.1908 | 4.52 | 0.0258 | 2.0214 | 0.0053 | −0.033 | 0.3617 |
| group_13965 | hypothetical protein | 0.153 | 0.038 | 4.11 | **0.0066** | 0.16685 | 3.59 | 0.0264 | 1.0196 | 0.0806 | 0.074 | 0.01 |
| group_9537 | hypothetical protein | 0.235 | 0.081 | 3.71 | **0.0009** | 0.13482 | 3.3 | **0.0035** | 1.0141 | 0.0216 | 0.049 | 0.025 |
| group_13443 | hypothetical protein | 0.224 | 0.075 | 3.81 | **0.001** | 0.16685 | 3.23 | **0.005** | 1.0762 | 0.0178 | 0.06 | **0.0083** |
| group_7606 | hypothetical protein | 0.141 | 0.025 | 6.26 | **0.0021** | 0.10991 | 3.07 | 0.0916 | 1.106 | 0.0975 | 0.097 | 0.0001 |
| dgcE | hypothetical protein | 0.341 | 0.138 | 3.37 | **0.0003** | 0.07663 | 3.06 | **0.0016** | 0.9197 | 0.0133 | 0.048 | **0.0083** |
| aqpZ_2 | aquaporin Z | 0.247 | 0.094 | 3.96 | **0.0005** | 0.16685 | 2.97 | **0.0085** | 0.923 | 0.0385 | 0.074 | **0.0001** |
| group_2409 | ISL3 family transposase ISEc38 | 0.259 | 0.106 | 3.07 | **0.002** | 0.1908 | 2.67 | **0.0083** | 0.803 | 0.0487 | 0.05 | **0.01** |
| pdeC_2 | putative cyclic di-GMP phosphodiesterase PdeC | 0.365 | 0.169 | 2.88 | **0.0008** | 0.10991 | 2.66 | **0.0026** | 0.7709 | 0.0296 | 0.041 | 0.015 |
| sul1 | sulfonamide-resistant dihydropteroate synthase Sul1 | 0.141 | 0.031 | 4.97 | **0.0037** | 0.17478 | 2.64 | 0.0939 | 1.102 | 0.0994 | 0.12 | 0.0001 |
| qacEΔ1 | quaternary ammonium compound efflux SMR transporter QacE Δ 1 | 0.153 | 0.031 | 5.46 | **0.0019** | 0.12853 | 2.63 | 0.1011 | 1.1732 | 0.0745 | 0.107 | 0.0001 |
| arsD_1 | arsenical resistance operon trans-acting repressor ArsD | 0.318 | 0.144 | 3.19 | **0.0006** | 0.17478 | 2.62 | **0.0043** | 0.7263 | 0.0563 | 0.068 | **0.0001** |
| group_774 | hypothetical protein | 0.459 | 0.25 | 2.55 | **0.0013** | 0.1391 | 2.55 | **0.0016** | 0.8601 | **0.0088** | 0.042 | **0.0033** |
| group_7231 | hypothetical protein | 0.306 | 0.138 | 2.69 | **0.0032** | 0.16685 | 2.54 | **0.0071** | 0.7807 | 0.0373 | 0.026 | 0.1733 |
| group_9530 | hypothetical protein | 0.459 | 0.238 | 2.64 | **0.0009** | 0.08496 | 2.38 | **0.0037** | 0.8449 | **0.0094** | 0.015 | 0.3533 |
| ltrA | hypothetical protein | 0.424 | 0.225 | 2.57 | **0.0013** | 0.16683 | 2.33 | **0.0051** | 0.7495 | 0.0229 | 0.037 | 0.015 |
| group_9528 | hypothetical protein | 0.353 | 0.163 | 2.87 | **0.001** | 0.12853 | 2.31 | 0.0134 | 0.7418 | 0.0384 | 0.032 | 0.0917 |
| group_622 | hypothetical protein | 0.482 | 0.269 | 2.71 | **0.0006** | 0.12566 | 2.24 | **0.0063** | 0.9086 | **0.0049** | 0.015 | 0.3683 |
| group_4187 | Antirestriction protein KlcA | 0.424 | 0.219 | 2.54 | **0.0017** | 0.12566 | 2.24 | **0.0092** | 0.7747 | 0.0191 | 0.021 | 0.1983 |
| umuD_2 | Protein UmuD | 0.6 | 0.369 | 2.42 | **0.0016** | 0.10991 | 2.1 | 0.0114 | 0.8272 | 0.0098 | 0.006 | 0.7 |
| umuC_2 | Protein UmuC | 0.682 | 0.456 | 2.52 | **0.0012** | 0.10991 | 2.09 | 0.0141 | 0.8239 | 0.0107 | 0.006 | 0.71 |
| group_2513 | hypothetical protein | 0.071 | 0.225 | 0.26 | **0.0038** | 0.16685 | 0.25 | **0.0031** | −1.2577 | 0.0119 | −0.02 | 0.3617 |

*Loci have identical presence/absence patterns across all sequenced isolates and are clustered for analysis.
**Bold text indicates significant results (unadjusted Q-value ≤ 0.2, P-value ≤ 0.01 after adjustment for clinical variables, and direct and/or indirect effect P-value ≤ 0.01).

Next, we aimed to adjust for the effects of phylogeny on the association between each group of genes and infection. We used treeWAS to test for the confounding potential of clonality and recombination in the population structure by simulating a null genetic dataset across the phylogenetic tree[29]. The output of treeWAS is three complimentary scores: The Terminal score counts the four possible combinations of genotype and phenotype without regard to phylogeny, whereas the Simultaneous and Subsequent scores account for population structure[29] in measuring the association between genotype and phenotype. Of the 28 infection-associated genes after adjustment (Table 2), 23 genes were also significant (P ≤ 0.01) by the Terminal score; group_774, group_9530, ltrA, group_622, group_4187, and group_2513 were not (Supplementary Fig. S8, Supplementary Table 1). Other than group_2513, 27 genes were significantly associated with

infection by either the Simultaneous or Subsequent score, or both (Supplementary Fig. S8). For subsequent analyses, we focused on these 27 genes associated with infection both after adjustment for clinical variables and after at least one test of independence from bacterial phylogeny.

## AMR genes are associated with infection independent of prior exposure to antibiotics

The identification of multiple AMR genes associated with infection after adjustment for exposure to high-risk antibiotics was unexpected. To investigate the association between AMR genes and infection further, we used two independent approaches. First, we identified classes of AMR genes as defined by Kleborate[25] in our strain set. Overall, the prevalence of many of these AMR determinants was low (<25%).

**Table 3 | *Klebsiella* genotype association with infection**

| Genotype | Case frequency | Control frequency | Unadjusted | | Adjusted for clinical variables | |
|---|---|---|---|---|---|---|
| | | | Odds ratio | P-value | Odds ratio | P-value |
| **Virulence determinants** | | | | | | |
| *ybt* | 0.207 | 0.133 | 1.7 | 0.144 | 1.78 | 0.155 |
| *col* | 0.024 | 0.007 | 3.72 | 0.286 | 0.629 | 0.709 |
| *iuc* | 0.024 | 0.007 | 3.72 | 0.286 | 0.436 | 0.507 |
| *iro* | 0.012 | 0.007 | 1.84 | 0.668 | 0.476 | 0.602 |
| *rmpA* | 0.024 | 0.007 | 3.72 | 0.286 | 0.436 | 0.507 |
| *rmpA2* | 0.012 | 0.007 | 1.84 | 0.668 | 0.937 | 0.964 |
| **AMR determinants*** | | | | | | |
| AGly acquired (aminoglycosides) | 0.268 | 0.067 | 5.133 | **7.01E-05** | 3.77 | **0.00427** |
| Flq (fluoroquinolones) | 0.220 | 0.040 | 6.75 | **1.14E-04** | 6.69 | **0.000507** |
| MLS (macrolides) | 0.098 | 0.013 | 8.0 | **0.010** | 6.41 | 0.0649 |
| Phe (phenicols) | 0.171 | 0.027 | 7.51 | **0.001** | 6.35 | **0.00332** |
| Rif (rifampin) | 0.012 | 0.000 | N/A | N/A | N/A | N/A |
| Sul (sulfonamides) | 0.268 | 0.047 | 7.49 | **1.22E-05** | 5.19 | **0.00226** |
| Tet (tetracyclines) | 0.183 | 0.100 | 2.01 | 0.076 | 1.88 | 0.128 |
| Tmt (trimethoprim) | 0.244 | 0.053 | 5.73 | **8.86E-05** | 4.28 | **0.00285** |
| Omp mutation | 0.037 | 0.020 | 1.86 | 0.453 | 1.55 | 0.676 |
| Bla (β-lactamase that is "other") | 0.573 | 0.520 | 1.24 | 0.438 | 1.27 | 0.379 |
| Bla Carb (carbapenemase) | 0.061 | 0.007 | 9.68 | 0.040 | 36.2 | **0.00151** |
| Bla ESBL (extended-spectrum β-lactamases) | 0.207 | 0.087 | 2.76 | 0.011 | 2.94 | 0.0116 |
| Bla broad (broad-spectrum β-lactamases) | 0.354 | 0.347 | 1.03 | 0.915 | 0.902 | 0.734 |
| Bla broad inhR (broad-spectrum β-lactamases with resistance to β-lactamase inhibitors) | 0.110 | 0.107 | 1.03 | 0.942 | 1.05 | 0.922 |

*Kleborate[25] predicted AMR genotypes absent in our dataset are not shown.
Bold text indicates significant results (P-value ≤ 0.01).

Consistent with results from Scoary, AMR determinants for aminoglycosides, fluoroquinolones, phenicols, sulfonamides, and trimethoprim were significantly associated with infection after adjustment. In addition, carbapenamase genes were also significantly associated with infection, and extended-spectrum β-lactamases (ESBL) approached significance (*P* = 0.011; Table 3). Notably, one case rectal isolate (Kp10372, Supplementary Data 4) displayed a hypervirulent (ST23, *ybt*+, *col*+, *iuc*+, *rmpA/A2*+), MDR (KPC-3) genotype. The Kleborate Resistance Score[25] was not significantly different between cases and controls (Case resistance score = 0.2 ± 0.483, Control resistance score = 0.163 ± 0.418, *P* = 0.528, Student's *t*-test); however, this score is heavily weighted by the presence of carbapenemases in the context of colistin resistance, the latter of which was not detected in our dataset[25]. Next, we assessed phenotypic resistance across four antibiotic classes, using an agar dilution method with antibiotic concentrations at the Clinical & Laboratory Standards Institute breakpoint for resistance[30]. Screen results were confirmed for 12 randomly selected *Klebsiella* isolates using the broth microdilution method. Overall, phenotypic AMR rates were low (<25%). Phenotypic resistance to chloramphenicol and gentamicin did not correlate well to genotypic resistance and was not associated with infection (Supplementary Table 2). Conversely, phenotypic resistance to ciprofloxacin, cefazolin, cefuroxime, and cefepime was associated with infection following adjustment for clinical variables (Supplementary Table 2).

Given the consistent association of AMR genes and phenotypes despite adjustment for exposure to high-risk antibiotics as a group, we evaluated whether their association with infection is due to exposure to specific cognate antibiotics. For example, *aac(6')-lb-cr5* may be associated with exposure to aminoglycosides, a class that was not included in adjustment. Interestingly, prior antibiotic exposure by specific classes was not associated with the presence of most infection-

associated genes in unadjusted analysis (Supplementary Fig. S7). Next, we explicitly adjusted for each class of antibiotic, in addition to clinical variables by IPTW, and assessed the association of each gene with infection. Even after individual adjustment for 18 classes of antibiotics (no patients had a prior exposure to daptomycin or polymyxin), all AMR genes remained significantly associated with infection (Supplementary Fig. S9). These data suggest an alternative explanation exists for the association between infection and AMR genes.

**Gut relative abundance mediates association between certain genes and *Klebsiella* infection**

We previously identified that high *Klebsiella* relative abundance in the gut is associated with infection[11], and that genes associated with infection can enhance fitness in the gut[23]. Therefore, we hypothesized that the *Klebsiella* genes associated with infection may act by enabling gut dominance that in turn increases the risk of infection. *Klebsiella* relative abundance measurements from the rectal swabs most proximal to infection for cases and matched controls were available for 233 of 245 sequenced rectal isolates[11]. Reanalysis of these data was consistent with that of the original study, wherein cases had significantly higher *Klebsiella* relative abundance in the gut compared to controls (Fig. 3a). Mean relative abundance was not significantly different between species (Supplementary Fig. S10a). Abundance was higher in patients exposed to high-risk antibiotics, and when stratified by antibiotics exposure the abundance in cases was significantly higher than controls (Supplementary Fig. S10b, c). Analysis of *Klebsiella* gut relative abundance based on gene presence/absence (rather than case status) identified 11 of the 27 infection-associated genes associated with increased relative abundance (Fig. 3b), whereas the other 16 infection-associated genes were not associated with increased relative abundance (Supplementary Fig. S10d).

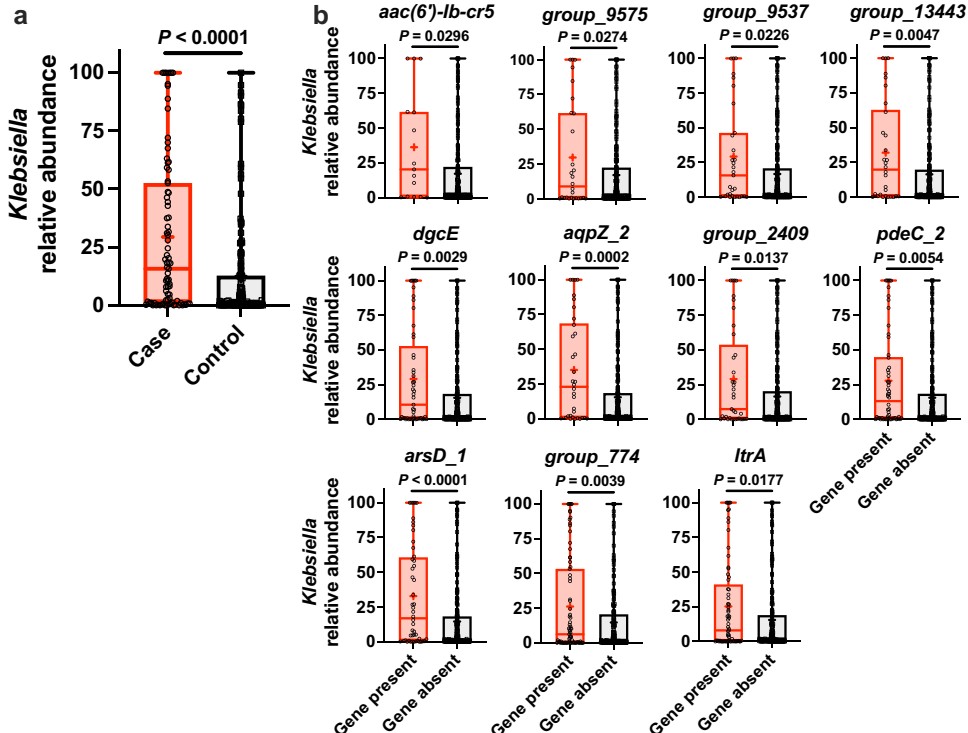

**Fig. 3 | Specific infection-associated genes are associated with increased *Klebsiella* gut relative abundance. a** Comparison of *Klebsiella* gut relative abundance in patient rectal swabs between cases (*n* = 83) and controls (*n* = 149, mean displayed, ****P*-value ≤ 0.00005, two-sided Student's *t*-test) and **b** in patient rectal swabs stratified by gene presence/absence (25% percentile and 75% percentile bound the box, median displayed as line, mean displayed as plus sign, and range displayed as whiskers, *P*-value ≤ 0.05, **P*-value ≤ 0.005, ***P*-value ≤ 0.0005, ****P*-value ≤ 0.00005, two-sided Student's *t*-test). Each data point represents a single patient rectal swab. Source data are provided as a Source Data file. Sample sizes for *aac(6')-Ib-cr5* present *n* = 15, *aac(6')-Ib-cr5* absent *n* = 217, *group_9575* present *n* = 28, *group_9575* absent *n* = 204, *group_9537* present *n* = 31, *group_9537* absent *n* = 201, *group_13443* present *n* = 30, *group_13443* absent *n* = 202, *dgcE* present *n* = 49, *dgcE* absent *n* = 183, *aqpZ_2* present *n* = 34, *aqpZ_2* absent *n* = 198, *group_2409* present *n* = 37, *group_2409* absent *n* = 195, *pdeC_2* present *n* = 54, *pdeC_2* absent *n* = 178, *arsD_1* present *n* = 48, *arsD_1* absent *n* = 184, *group_774* present *n* = 75, *group_774* absent *n* = 157, *ltrA* present *n* = 69, and *ltrA* absent *n* = 163.

To test the hypothesis that infection-associated genes may act through gut dominance we undertook a mediation analysis, a statistical approach that estimates the strength of both the direct associations between each gene and the outcome of infection and indirect association through gut relative abundance. For seven gene units, the association with infection was mediated, at least in part, by gut abundance (Table 2; indirect *P*-value, indicated in bold). The AMR gene *aac(6')-Ib-cr5* had both a strong direct association with infection and an indirect mediation by abundance (*P*-value ≤ 0.01). The hypothetical protein *group_774* also had both significant direct and indirect effects. The remaining five genes (hypothetical protein *group_13443*, *dgcE*, *aqpZ_2*, transposase *group_2409*, and *arsD_1*) had only significant indirect effects, indicating that their association with infection is mediated by gut relative abundance. Collectively, these findings begin to suggest biological roles for *Klebsiella* genes associated with infection, in which some genes may act through high gut relative abundance, but others likely have separate functions.

**Most infection-associated genes are carried on plasmids**
Given the diversity of functions and frequencies of the infection-associated genes, we sought to determine the genetic context of these genes as part of the accessory genome. Based on nucleotide homology, many of these genes have been previously detected on plasmids[31–33]. Consistent with plasmid carriage, certain infection-associated genes, including both AMR and non-AMR genes, are highly associated with one another (Supplementary Fig. S7) and appear to be linked (Fig. 2b). Furthermore, we detected the loss or gain of groups of infection-associated genes between some case clinical and rectal pairs, rather than individual genes (strata 211, 214, 701, 713,

716, Supplementary Fig. S3f). To determine the association between plasmids and infection-associated genes, we used the PlasmidFinder tool to detect replicons in the rectal isolates in our dataset[34]. We detected a replicon in 163 of 245 isolates (66.5%, 72.94% of case rectal isolates, 63.13% of control isolates), with a median of 2 replicons per isolate (Supplementary Data 5, range = 0–10 replicons). There was no significant difference between the number of replicons per isolate between cases and controls (Supplementary Fig. S11a). The three most common replicons were Col(pHAD28), IncFIB(K), and IncFII(K), and these replicons were distributed across the core genome phylogeny (Supplementary Fig. S11b); however, no replicon was significantly associated with infection (Supplementary Table 3).

To determine whether infection-associated genes are located on plasmids or the chromosome, we performed long-read sequencing and hybrid assembly with existing short read data on 10 isolates from cases of infection (Figs. 4a–c, S12). All of these isolates contained plasmids with infection-associated genes, and no two plasmids containing infection-associated genes were identical in terms of size, organization and gene content. At one extreme, the rectal case isolate Kp11407 contained a 208,703 bp hybrid IncFIB(K)/IncFII(K) plasmid encoding all 18 infection-associated loci present in this isolate (Fig. 4a). This plasmid closely aligns to the conjugative plasmid pUUH239.2 (CP002474.1), which was linked to a nosocomial *K. pneumoniae* outbreak[35]. Notably, this plasmid also contains several heavy metal resistance operons, including *ars*, *pco*, and *sil*[35], which may enhance resistance to innate immune responses[36]. A second case rectal isolate, Kp5854, encoded 23 of 27 infection-associated loci on a 247,093 bp hybrid IncFIB(K)/IncFII(K) plasmid (Fig. 4b). This plasmid closely aligns to pF16KP0070-1 (CP052586.1), which also encodes *ars*, *pco*, and *sil*.

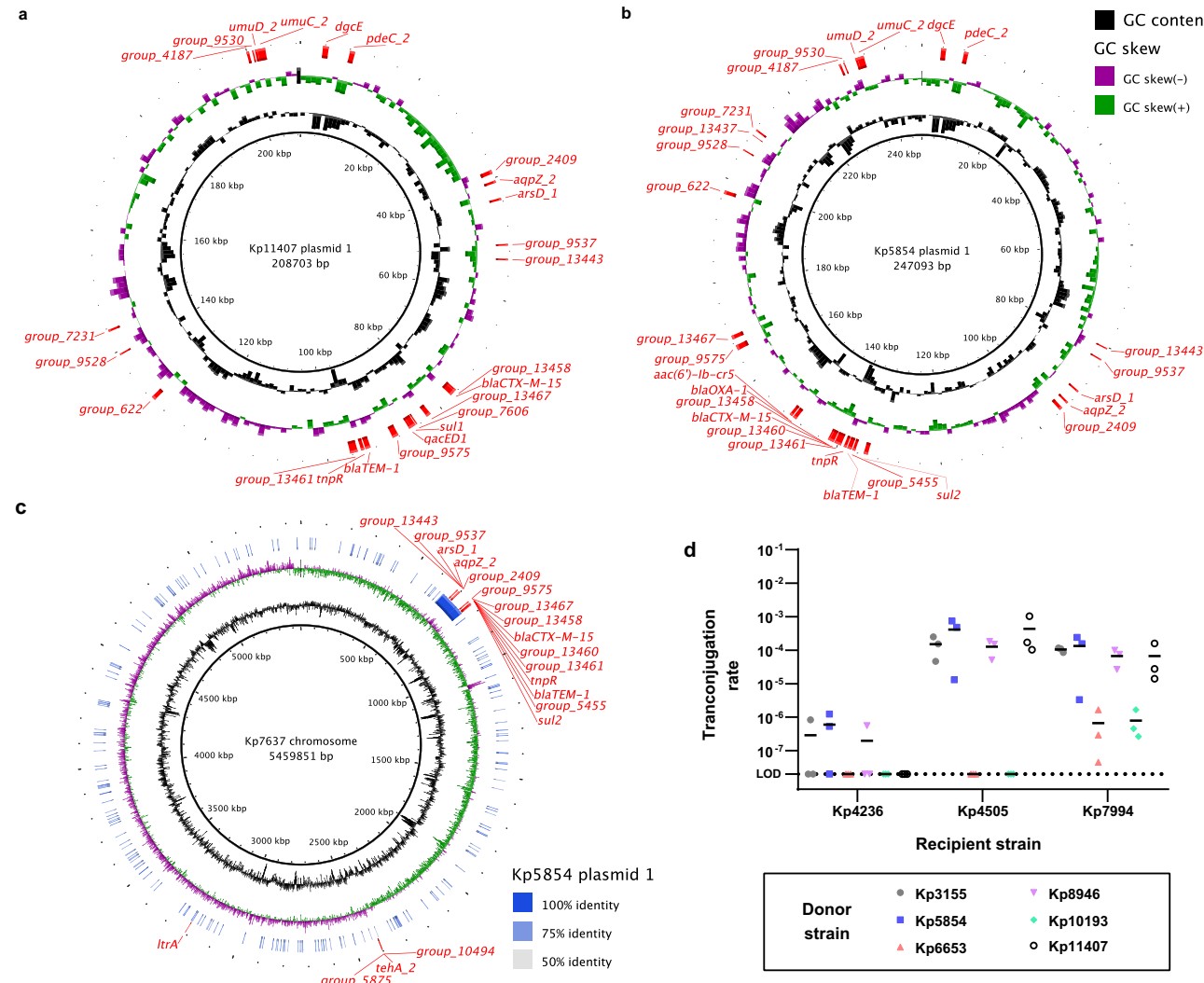

**Fig. 4 | Infection-associated genes are present on large, conjugative plasmids. a** Plasmid map of Kp11407 plasmid 1. **b** Plasmid map of Kp5854 plasmid 1. **c** Alignment of Kp7637 chromosome and Kp5854 plasmid 1. **d** Transconjugation rates using Kp3155, Kp5854, Kp6653, Kp8946, Kp10193, and Kp11407 as donor strains to Kp4236, Kp4505, and Kp7994, which are *K. quasipneumoniae*, *K. pneumoniae*, and *K. variicola*, respectively, as recipient strains (mean displayed, N.D. = none detected). For **d**, each data point represents a single biological replicate. Source data are provided as a Source Data file.

Interestingly, in the corresponding clinical isolate, this plasmid appears to have integrated into the chromosome with the loss of some infection-associated genes (Fig. 4c; stratum 713, Supplementary Fig. S3g). These isolates also contained four additional infection-associated genes that mapped to the chromosome. The number of plasmid-encoded, infection-associated genes varied widely between plasmids (Supplementary Fig. S12). In contrast to Kp11407 and Kp5854, Kp11162 contained one plasmid with two infection-associated genes and a second plasmid with four (Supplementary Fig. S12g, h). Across these isolates 23 of all 27 infection-associated genes were found on at least one plasmid (Supplementary Fig. S12). Only *group_774* and the *tehA_2* gene cluster were not found on at least one plasmid. This suggests that these large plasmids facilitate acquisition of infection-associated loci and are likely drivers of the association between many genes and infection.

### Infection-associated genes are transmissible by plasmids

To determine if infection-associated genes could be horizontally transmitted, we performed transconjugation assays between rectal isolates. Kp3155, Kp5854, Kp6653, Kp8946, Kp10193, and Kp11407 were selected as donor strains, as these *K. pneumoniae* case rectal isolates contained various infection-associated genes, including

$bla_{CTX-M-15}$ and *dgcE*, as well as the IncFIB(K) replicon. Three cefuroxime-sensitive control isolates lacking these genes, *K. pneumoniae* (Kp4505), *K. variicola* (Kp7994), and *K. quasipneumoniae* (Kp4236), were selected as recipient strains. Kp7994 and Kp4236 lack plasmid replicons, and Kp4505 contains an IncFIB(K) replicon. Conjugation to Kp4236 was limited, with inconsistent or lack of transfer across donor strains. In contrast, conjugation was readily detected from donors Kp3155, Kp5854, Kp8946, Kp11407 to recipient Kp4505, and from all donors to Kp7994 (Fig. 4d). All transconjugants were cefuroxime resistant, indicating the transfer of $bla_{CTX-M-15}$, and were PCR-positive for *dgcE*. Therefore, we concluded that these large plasmids are conjugative, resulting in the transfer of infection-associated genes.

### Validation of infection-associated genes in a geographically distinct patient cohort

Finally, we aimed to validate our findings across multiple geographical locations. To this end, we screened a cohort of *Klebsiella* colonized patients from a single academic medical center in Maryland, USA for six infection-associated genes: *aac(6')-Ib-cr5*, $bla_{CTX-M-15}$, *group_13461*, *group_13467*, *dgcE*, and *aqpZ_2*. These genes were selected because they were representative of the many facets of our findings,

**Table 4 | Gene association with positive clinical cultures in a geographically independent cohort of *Klebsiella* colonized patients**

| Gene | Sample | Frequency | Odds ratio | 95% CI | *P*-value |
|---|---|---|---|---|---|
| *aac(6')-Ib-cr5* | Case clinical | 0.167 | 4.88 | 1.96 to 11.1 | **0.0009** |
| | Case rectal | 0.115 | 3.18 | 1.22 to 8.41 | **0.0276** |
| | Control rectal | 0.039 | | | |
| *dgcE* | Case clinical | 0.481 | 2.28 | 1.29 to 3.99 | **0.0074** |
| | Case rectal | 0.5 | 2.46 | 1.37 to 4.41 | **0.0039** |
| | Control rectal | 0.289 | | | |
| *group_13461* | Case clinical | 0.204 | 4.55 | 2.06 to 9.6 | **0.0005** |
| | Case rectal | 0.192 | 4.23 | 1.93 to 9.24 | **0.0042** |
| | Control rectal | 0.053 | | | |
| *group_13467* | Case clinical | 0.296 | 3.81 | 1.94 to 7.29 | **0.0002** |
| | Case rectal | 0.25 | 3.02 | 1.48 to 6.1 | **0.0043** |
| | Control rectal | 0.1 | | | |
| *bla*$_{CTX-M-15}$ | Case clinical | 0.185 | 4.94 | 2.19 to 11.4 | **0.0005** |
| | Case rectal | 0.173 | 4.55 | 1.83 to 10.9 | **0.0013** |
| | Control rectal | 0.044 | | | |
| *aqpZ_2* | Case clinical | 0.296 | 1.58 | 0.83 to 2.93 | 0.164 |
| | Case rectal | 0.327 | 1.82 | 0.98 to 3.38 | 0.076 |
| | Control rectal | 0.213 | | | |

Bold text indicates significant results (*P*-value ≤ 0.05).

encapsulating known AMR genes (*aac(6')-Ib-cr5*, *bla*$_{CTX-M-15}$), MGEs (*group_13461*, *group_13467*), and those where the relationship between gene presence and infection was (*aac(6')-Ib-cr5*, *dgcE*, *aqpZ_2*) and was not (*bla*$_{CTX-M-15}$, *group_13461*, *group_13467*) significantly mediated by *Klebsiella* gut relative abundance. Moreover, these genes are found on conjugative plasmids (Fig. 4). This second cohort consisted of 55 cases of *Klebsiella* infection and 432 asymptomatically colonized control patients. Since *wzi* concordance data was not available, we screened both case isolates and a rectal isolate from each patient for each gene. Rectal isolates were not available for three infected patients and a clinical isolate was not available for one patient. In total, 538 *Klebsiella* isolates (54 clinical isolates, 52 case rectal isolates, 432 control rectal isolates) were screened for the presence or absence of target genes using a multiplex RT-PCR assay. As an internal control for *Klebsiella* detection, we included a previously validated assay for the *fiu* gene[11]. Based on either clinical or rectal isolates from cases, we observed a significant association with infection for all genes except *aqpZ_2* (Table 4). Overall, 82.4% (42/51 of available pairs) of clinical-rectal pairs returned concordant RT-PCR results, which was highly consistent with our previous study using *wzi* sequencing to determine concordance[10]. As in the primary cohort (Supplementary Fig. S7), the presence of these genes in case rectal isolates were associated with one another (Supplementary Fig. S13), suggesting that these genes in the Maryland isolates may also be linked through conjugative plasmids. The strongest associations were between *dgcE* and *apqZ_2* and between *aac(6')-Ib-cr5*, *bla*$_{CTX-M-15}$, *group_13461* and *group_13467*. Overall, five of six genes were reproducibly associated with case status in colonized patients in two geographically-distinct cohorts.

## Discussion

In this study, we have identified novel *Klebsiella* genes associated with infection in colonized patients. The findings from this diverse collection of *Klebsiella* isolates, carefully adjusted for patient risk factors and phylogeny, and validated in a second geographically distinct cohort, provides important insights into the broader dynamics of *Klebsiella* pathogenesis. Most of these genes were found across a diverse set of plasmids, suggesting that these genes are either direct mediators of fitness and virulence or indirect markers of large, conjugative plasmids

that increase infection risk as a unit. These genes can be separated into two categories: those whose association with infection is mediated through increased abundance in the gut, and those that have a direct association with infection. Genes whose effect on infection risk is not mediated through intestinal dominance may act at a subsequent step of pathogenesis. These genes associated with infection are readily mobilized within and across members of the *Klebsiella pneumoniae* species complex, and changes in infection-associated gene content and genetic context were detectable within colonized patients who progressed to infection. Collectively, this study provides a set of genetic signatures of infection that have potential for predicting risk of infection in patients colonized by diverse *Klebsiella*.

The majority of infection-associated loci in our dataset are present on plasmids, which can vary in size, gene content, copy number, and conjugative ability (Fig. 4a–c, Supplementary Fig. S12). This plasmid carriage creates genetic linkage between genes that are associated with infection, raising the possibility that some are direct effectors of fitness or virulence whereas others are robust markers for these effectors. In some instances, clear relationships exist between linked genes. For example, nucleotide cyclases are often present in tandem with cyclic nucleotide phosphodiesterases, thus the linkage of *dgcE* and *pdeC_2* is expected. *dgcE* and *pdeC_2* may regulate expression of type 1 and 3 pili through modulation of cyclic di-GMP levels. Expression of the type 1 pilus reduces virulence in the lung and spleen but enhances gut colonization in murine models[37,38]. Therefore, *dgcE* may enhance fitness in the gut while *pdeC_2* may enhance fitness in extra-intestinal sites. Likewise, the linkage of *bla*$_{CTX-M-15}$ and *group_13461* is expected since ISEcp1 transposases are known to mobilize and promote the expression of CTX-M-type β-lactamases[39,40]. Conversely, relationships between other linked genes, such as *arsD_1* and the various AMR genes, are more difficult to understand. It may be the case that these infection-associated genes act at different stages of infection or one or both are robust genetic markers for fitness genes elsewhere on the plasmid.

The strong association between multiple AMR genes and infection following adjustment for high-risk antibiotic exposure indicates that the relationship between antibiotic resistance and infection is complex in *Klebsiella*. Antibiotic resistance genes could be associated with

infection by enabling intestinal outgrowth and dominance after exposure to a cognate antibiotic. However, the ESBL determinants $bla_{CTX-M-15}$, $bla_{OXA-1}$, and $bla_{TEM-1}$ were associated with infection independent of antibiotic exposure and their association was not significantly mediated by gut relative abundance. Notably, antibiotic exposure rates are low (<10% for most classes) in our original study population[10]. Therefore, detection of these antibiotic resistance genes is more likely a marker of other infection-associated genes and/or plasmids, rather than conferring resistance to a cognate antibiotic. It may be the case that these strains acquired these genes earlier in their evolutionary history; however, prospective comparative genomics studies are necessary to pinpoint such events. Potential fitness benefits of AMR genes in the absence of antibiotics are intriguing, yet highly AMR *Klebsiella* STs are generally less virulent in experimental models than their hypervirulent counterparts, and specific mutations that confer AMR can have deleterious effects on *Klebsiella* fitness[41–44]. On the other hand, plasmid acquisition of AMR is less costly than chromosomal mutations[45]. Therefore, AMR genes may be linked to plasmid-encoded genes that enhance infection risk, and then complicate treatment of infections after they are initiated.

The identification of non-AMR genes associated with infection as mediated by gut dominance adds to our understanding of bacterial genotypes and the risk for infection in *Klebsiella* colonized patients. We previously demonstrated that the *ter* operon, annotated as a tellurite-resistance operon, is a microbiome-dependent gut fitness factor[23]. Similarly, the *arsD_1* gene identified in this study is part of an arsenic resistance (*ars*) operon. Tellurite and arsenic are both toxic metalloids, implying that the *ars* operon may be playing an analogous role to *ter*. Consistent with this, gut relative abundance mediates the association between *arsD_1* and infection (Table 2). Conversely, we did not find evidence that gut relative abundance mediates the association between the tellurite-resistance gene *tehA_2* gene and infection, although it is predicted to encode a different molecular function than *ter*. These data highlight the complex interplay between bacterial genotype and gut fitness, and the need to understand the role of colonization fitness to both assess infection risk and prophylactic decolonization therapies. The risk of *Klebsiella* disease in colonized patients may not warrant prophylactic antibiotic administration, especially in the context of rising rates of AMR; however, non-antibiotic options are currently being explored (reviewed in[46]). A better understanding of how bacterial pathogens, such as *Klebsiella*, successfully colonize and dominate the gut is needed to ensure the efficacy of any decolonization approach.

Although this study advances our understanding of infection in patients colonized by diverse *Klebsiella*, it is not without its limitations. First, this study is retrospective and relies on the availability of clinical data in the electronic medical record. Prospective studies with longer timeframes aimed at evaluating the association between genes identified in this study and infection would be useful for validating our findings. Second, many infection-associated genes are associated with one another and these genes are plasmid-borne. This finding is important; however, it complicates the evaluation of the causal relationship between individual genes and infection. Experimental studies are necessary to determine the causal relationship between each gene and disease. Additionally, we did not comprehensively assess the genomic architecture and HGT potential of our strains. It may be the case that only a subset of genes is frequently transferred between strains and/or only a subset of strains is able to easily donate and receive these genes. Notably, all plasmid donor strains that we tested were *K. pneumoniae* that encoded an IncFIB(K) replicon (Supplementary Data 5), therefore limiting our ability to assess the universality of the HGT phenotype. Further assessment of the *Klebsiella* plasmidome will be critical to fully interpret the role of the infection-associated genes identified in this study, as well as the characterization of cryptic plasmids that could be missed with replicon-based analysis such as

PlasmidFinder. Moreover, more complicated approaches such as shotgun metagenomics are necessary to contextualize plasmid borne genes in polymicrobial reservoirs of bacteria with pathogenic potential, such as the gut. In our combined approach across three species, the majority of our genomes were from *K. pneumoniae*. There may be *K. variicola* or *K. quasipneumoniae*-specific infection-associated genes that we were unable to detect. Third, the *Klebsiella* genes associated with infection likely differ by patient population. We found different genes associated with infection (*ter*, psicose utilization locus) in a hospital-wide comparative genomics study and experimentally validated them as encoding fitness factors[22,23]. That patient population may have different risk factors and stronger barriers to infection. Among intensive care and hematology/oncology patients, the genes associated with infection were highly reproducible in a geographically independent cohort. Finally, this study focused on gene presence in the accessory genome and does not consider the role of mutations within the core or accessory genomes.

This study provides insights into infection risk from diverse lineages of the *K. pneumoniae* species complex. The well-characterized virulence genes of hypervirulent clonal groups, many of which are plasmid-borne[25], were rare and not informative in assessing infection risk in this population. MDR isolates were more frequent representing six clonal groups, but overall, 85.7% (210/245) of the isolates in our collection were from non-hypervirulent, non-MDR CGs. Given the predominance of non-hypervirulent, or "classical" strains worldwide[25], efforts to understand the pathogenesis and enact preventative therapies is likely to have a substantial payoff. The varied combinations of infection-associated genes found across case isolates suggests that a diverse accessory genome acquired through HGT drives infection risk in *Klebsiella*. Certain combinations have become fixed; some hypervirulent *Klebsiella* clones have carried their plasmids for decades. However, new combinations can arise within a colonized patient (Fig. 4; Supplementary Fig. S12). Even in seemingly fixed plasmid-strain combinations, exchange has occurred leading to multiple instances of the convergence of virulence and MDR determinants[25]. In total, this study adds to the growing understanding of the *Klebsiella* accessory genome as a critical determinant of infection risk.

The insights from this study can be used to predict infection risk in colonized, hospitalized patients and trigger infection prevention interventions. We have previously identified certain patient risk factors associated with infection, and that gut dominance is associated with infection independently of these clinical factors[10,11]. Here we identified infection-associated genes independent of patient risk factors and that part of the association is mediated by gut dominance. We have also developed the tools to measure these risks, including a robust qPCR assay to measure intestinal dominance and the genotyping PCR assays implemented in this study. The fact that AMR genes and phenotypes are strongly associated with infection could also enable the use of standard culture approaches to detect these genes, and therefore, high-risk colonizing strains. It is possible that these approaches can be combined with queries of the electronic medical record for clinical variables to generate an integrated risk score for a colonized patient. In the near term, this score could be used to identify patients for trials of decolonization therapies. Eventually, integrated risk models could identify at-risk patients who could most benefit from effective prevention interventions.

## Methods

### Ethics statement
Patient enrollment and sample collection at the University of Michigan was approved by and performed in accordance with the Institutional Review Boards (IRB) of the University of Michigan Medical School (Study number HUM00123033). Patient enrollment and sample collection at Johns Hopkins University was approved and performed in accordance with the IRB of the Johns Hopkins

University (Study number IRB00129775). Patients were retrospectively enrolled after routine samples were taken in the course of clinical care. This study was performed with a waiver of informed consent since the research involves no more than minimal risk to the subjects, could not practicably be carried out without the waiver, and uses discarded samples. Patient enrollment was identical at both sites.

## Sample collection and strain selection

At Michigan Medicine, *Klebsiella* isolates were selected from a large cohort of *Klebsiella* colonized patients. Cohort identification, enrollment, clinical data extraction, chart review, and case definitions are described in detail elsewhere[10]. During the study period, patient rectal swabs were screened for the presence of *Klebsiella* by plating on MacConkey agar followed by taxonomic identification using MALDI TOF mass spectrometry. Up to three *Klebsiella* isolates per rectal swab, and any subsequent *Klebsiella* positive clinical cultures, were archived. Minimum inhibitory concentration data from case clinical isolates was extracted from the Michigan Medicine clinical microbiology lab. To identify concordant infections, *Klebsiella* rectal swab and case clinical isolates were subjected to *wzi* PCR and Sanger sequencing, and the *wzi* type was assigned by uploading the consensus *wzi* sequence to the BigsDB database (https://bigsdb.pasteur.fr). Each case was matched to two controls based on sex, age (±10 years), and date of rectal swab collection (±90 days). Several cases could only be matched to a control swab within 120 days ($n = 6$). Case matching was conducted in R, version 4.0.3.

At the Johns Hopkins University, adult patients admitted to medical or surgical ICUs are routinely screened for VRE rectal colonization on admission and weekly thereafter until ICU discharge. Residual Amies broth from rectal ESwabs (COPAN Diagnostics, Inc., Murrieta, California) were inoculated on a MacConkey agar plate using the WASP DT (COPAN) automated specimen processer, and isolates were archived as above.

## Whole-genome sequencing, assembly, and annotation

Archived *Klebsiella* isolates were cultured in Luria-Bertani (LB, Becton, Dickinson and Company, Franklin Lakes, NJ) broth at 37 °C with shaking for genomic DNA extraction. Genomic DNA was extracted using the DNeasy UltraClean Microbial Kit (Qiagen, Hilden, Germany) per manufacturer's instructions. Extracted genomic DNA was then checked for quality and purity and submitted to the University of Michigan Advanced Genomics Core, where it was sequenced on an Illumina NovaSeq using a 300 cycle S4 flow cell (Illumina, San Diego, USA) to generate approximately 6,000,000 150 bp paired-end reads. Resulting reads were trimmed using Trimmomatic v0.39[47] and assembled into scaffolds using SPAdes v3.13.0[48]. Genome assemblies within each stratum were compared using QUAST v5.0.2[49], wherein each genome assembly was compared to the case rectal isolate genome assembly. Genome annotation was performed using Prokka v1.14.6[50], and a pangenome was assembled using Roary v3.13.0[51], with settings "-i 90" and "-s." Finally, assembly genotyping was performed using Kleborate v1.0.0[25] and PlasmidFinder v2.0.1[34]. For long-read sequencing, genomic DNA was checked for quality and purity and submitted to the University of Michigan Advanced Genomics Core, where it was sequenced on a GridION X5 platform using a MinION flow cell (Oxford Nanopore Technology, Oxford, UK). Low-quality read ends were trimmed with NanoFilt v2.5.0[52] and assembled using Unicycler v0.4.8 using the "bold" setting[53]. The core gene alignment was created using Illumina assemblies with cognac using the default parameters[54]. The *wzi* gene sequences from each Illumina assemblies were identified by BLAST v2.9.0[55]. The BLAST output was parsed to extract the aligned sequence from each query, and the gene sequences were aligned with MAFFT v7.310. Approximate maximum likelihood trees were generated from the *wzi* alignment and cognac alignment

with fastTree v2.1.10[56]. Phylogenetic trees were visualized with the APE R package v5.3[57]. Rarefaction was performed using iNEXT v2.0.20[58], with setting "$q = 0$." All analyses were performed in R v4.0.5. Plasmid maps and corresponding alignments were constructed and visualized using BRIG v0.95[59]. All code developed for this manuscript is available at https://github.com/rdcrawford/Bachman_CU8.

## Phenotypic antibiotic screen

To screen for phenotypic antibiotic resistance, all *Klebsiella* rectal isolates were arrayed into 96-well plates and stored at −80 °C in 20% LB-glycerol. We were unable to locate one isolate in our freezer following WGS, thus it was omitted from phenotypic analysis. Then *Klebsiella* rectal isolates were sub-cultured in LB broth at 37 °C overnight and replica plated on antibiotic containing LB-agar. Antibiotic concentrations (36 mg/mL ampicillin, 32 mg/mL chloramphenicol, 16 mg/mL gentamicin, 1 mg/mL ciprofloxacin, 32 mg/mL cefazolin, 32 mg/mL cefuroxime, 16 mg/mL cefepime) were selected in accordance with CLSI criteria for resistance. All assays were performed in triplicate.

## Transconjugation assay

Single colonies of Kp4236, Kp4505, Kp7994 (recipient strains; chloramphenicol resistant), and Kp3155, Kp5854, Kp6653, Kp8946, Kp10193, Kp11407 (donor strains; cefuroxime resistant) were sub-cultured overnight in 1 mL of LB broth at 37 °C with shaking. The next day, overnight cultures were harvested by centrifugation and resuspended in 1 mL sterile phosphate-buffered saline (PBS). 50 μL of Kp4236, Kp4505, or Kp7994 resuspension was mixed with 50 μL of each donor strain resuspension and spotted on LB-agar. 50 μL of unmixed resuspension of each strain was spotted in parallel. Bacterial spots were grown overnight at 37 °C, then harvested into 1 mL sterile PBS by scraping and serially plated on differential growth media. Transconjugation rates were calculated by quantifying the bacterial density of dual-resistant colonies divided by the total density of the recipient strain. Transconjugates were confirmed by qPCR targeting the *dcgE* gene using unmixed strains as a reference (assay details below). All assays were performed in triplicate, and 3 transconjugate colonies and 3 unmixed recipient and donor colonies were selected per assay for PCR confirmation.

## RT-PCR design and assay

To screen for the presence of *aac(6′)-Ib-cr5*, *bla*$_{CTX-M-15}$, *group_13461*, *group_13467*, *dgcE*, and *aqpZ_2*, multiplex real-time PCR assays were designed using PanelPlex (DNA Software, Ann Arbor, USA) and combined with a previously validated *Klebsiella*-specific *fiu* assay[11]. All primer and probe sequences can be found in Supplementary Table 4. Clinical and rectal isolates were arrayed and sub-cultured in 150 μL LB overnight at 37 °C. To extract genomic DNA, overnight cultures were harvested by centrifugation and resuspended in 100 μL of PCR-grade water, boiled at 95 °C for 5 min, and pelleted. Then, 5 μL of the supernatant was used as a PCR template. Primer (Integrated DNA Technologies, Coralville, USA) and probe (Thermo Fisher Scientific, Waltham, USA) concentrations were 400 nM and 200 nM, respectively[11]. Real-time PCR was performed using a QuantStudio 3 real-time thermocycler (Thermo Fisher Scientific, Waltham, USA) using the following protocol: 50 °C for 2 min, 95 °C for 15 min, then 40 cycles of 94 °C for 1 min and 60 °C for 1 min. All assays were performed in duplicate unless replicates conflicted, wherein a third replicate was performed, and all isolate arrays included 3 positive controls and 3 randomly dispersed negative controls. The target gene was assigned as present in an isolate if a reaction was (1) positive for *fiu*, (2) amplified within 2 Ct of the range of the positive control Cts, and (3) results were concordant between all replicates.

## Statistical analysis

For clinical variable comparisons, odds ratios and associated 95% confidence intervals were calculated using conditional logistic

regression in R (version 3.6.3) and a *P*-value ≤ 0.05 was considered statically significant. For infection-associated gene discovery, a *P*-value ≤ 0.05 and *Q*-value ≤ 0.2 was used as a cutoff for further exploration. For these analyses, Scoary v1.6.16[26] was used with the "–collapse" and "-n" settings with our custom core genome phylogeny, limited to genes that were 5–95% frequent in the dataset. For analyses considering population structure, treeWAS v1.1[29] was used with the phen.type = "discrete" setting and our custom core genome phylogeny. For gene-gene associations, odds ratios were calculated using Fishers exact test in R Studio v1.2.5001. To account for the inflated type I error rate due to the large number of comparisons, a *P*-value ≤ 0.01 was considered statistically significant for subsequent clinical variable adjustment, mediation, population structure, and genotyping analyses.

Next, we considered adjustment of clinical variables. Inverse probability of treatment weighting method was used for this purpose[60]. The analysis is performed separately for each gene. In the propensity score model, a logistic regression was used with the clinical variables as predictors and binary gene presence/absence as the outcome. The predicted probabilities estimated from the regression model were used to compute weights and then included in the outcome model. The outcome model is a logistic regression with *Klebsiella* infection as the outcome and gene (presence/absence) as the predictor, while incorporating the matched nature of the data. Following IPTW analysis, we assessed if each gene was associated with infection after individually adjusting for exposure to each class of antibiotics. We repeated the described IPTW analysis using the same weights, controlling for each prior exposure as a covariate in the outcome model one at a time. The "survey" package (v4.1-1) in R was used to obtain the robust standard error.

We then conducted a mediation analysis on each gene that was significantly associated with infection to explore if the relationship between gene presence/absence and infection is mediated through abundance. Specifically, we assessed (1) the direct effect defined as the effect of gene presence on infection, adjusting for abundance as a continuous variable and clinical factors, (2) the indirect effect of the gene on infection mediated through abundance after adjusting clinical factors. To account for the matched case-control design, a generalized linear mixed model was used with a random effect for the strata defining matched case-controls. A significant mediation (i.e., indirect) effect indicates evidence that the effect of the gene on infection is mediated through abundance, and direct effects were also reported. The *P*-values for these effects were assessed to identify genes to investigate further. We conducted the analyses using the 'mediation' (v4.5.0) and 'lme4' packages (v1.1-29)[61,62] in R v3.6.3.

For in vitro experiments, one-sample *t*-test, unpaired Student's *t*-tests, and ANOVA followed by indicated post-hoc test was used to determine significant differences between groups following log transformation of experimental values. For in vitro experiments, a *P*-value ≤ 0.05 was considered statistically significant. Experimental replicates represent biological replicates. Statistical testing was two-sided in all circumstances. Statistical analysis of in vitro experiments was performed using Prism 8 (GraphPad Software, La Jolla, CA).

### Reporting summary
Further information on research design is available in the Nature Research Reporting Summary linked to this article.

## Data availability
The sequencing data generated in this study have been deposited in the Sequence Read Archive (SRA) database under accession code PRJNA789565. The human subjects data are available under restricted access for the protection of the rights and welfare of human research subjects. Deidentified human data are available under restricted access

and can be obtained from MAB within 1 year upon request, pending approval from the University of Michigan Institutional Review Board. The remaining data generated in this study are provided in the Supplementary Information/Source Data file. Source data are provided with this paper.

## Code availability
All code developed for this manuscript is available at https://github.com/rdcrawford/Bachman_CU8.

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

## Acknowledgements

The authors would like to thank the University of Michigan Advanced Genomics Core of the University of Michigan Medical School's Biomedical Research Core Facilities for their assistance with whole-genome sequencing. Additionally, we acknowledge support from the Bioinformatics Core of the University of Michigan Medical School's Biomedical Research Core Facilities. Figure 1a was created with BioRender.com. This work was supported by funding from National Institution of Health (https://www.nih.gov/) grants R01AI125307 to MAB and 1R01AI148259-01 to ESS. J.V. was supported by the Postdoctoral Translational Scholar Program (NIH UL1TR002240). R.C. was supported by the University of Michigan bioinformatics training grant (NIH T32GM070449). L.U. was supported by the Molecular Mechanisms in Microbial Pathogenesis Training Program (T32 AI007528). C.L.H. was supported by the Lung Immunopathology Training Program (T32HL007517). The funders had no role in study design, data collection and analysis, decision to publish, or preparation of the manuscript.

## Author contributions

Conceptualization: J.V., L.Z., P.J.S., P.D.T., K.R., K.S.K., M.A.B. Methodology: E.K.R., R.C., E.S.S., L.Z., K.R., M.A.B. Investigation: J.V., E.K.R., L.U., A.P., R.C., S.M., C.L.H., Y.S., A.T. Visualization: J.V., R.C. Funding acquisition: M.A.B. Project administration: M.A.B. Supervision: MAB. Writing—original draft: J.V., E.R., L.U., S.M., M.A.B. Writing—review & editing: J.V., E.K.R., L.U., A.P., R.C., S.M., C.L.H., Y.S., A.T., E.S.S., L.Z., P.J.S., P.D.T., K.R., K.S.K., M.A.B.

## Competing interests

K.R. is supported in part from an investigator-initiated grant from Merck & Co, Inc. and has consulted for Bio-K + International, Inc., Roche Molecular Systems, Inc., and Seres Therapeutics. All other authors declare no competing interests.
