## [Peer Review file · Nature Communications]

REVIEWER COMMENTS

Reviewer #1 (Remarks to the Author):

This study led by Jay Vornhagen of Michael Bachman's group at the University of Michigan examines *Klebsiella* isolates from colonized patients and performs a case-control comparative genomic analysis between those that went on to cause distant infection versus those that did not. Ultimately, the group identified 27 genes in the accessory genome that were associated with progression to infection. A subset of these genes were associated with gut abundance and a subset were not, suggesting they likely play a role in fitness at another point in pathogenesis. Interestingly, many of these genes were plasmid-borne.

This extremely-well written manuscript tackles an increasingly important issue in our patients. MDR *Klebsiella* strains are considered urgent or serious threats by the CDC, and *Klebsiella* is becoming one of our most troublesome nosocomial pathogens. Despite its importance, very little is understood about how this opportunist causes infection. Specifically, when a large fraction of society is colonized with *Klebsiella* (and an even larger fraction of hospitalized patients), how is it determined who goes on to suffer infection with *Klebsiella*? This group previously studied host factors that may make a patient more susceptible for progression to *Klebsiella* infection. Here, through WGS and a series of statistical models, they identify organism-specific factors that correlate with progression from colonization to infection. Overall, the work herein is well-described, well-controlled, transparent, rigorous, and important. While this study did not identify a "smoking gun" if you will, it did identify several *Klebsiella* factors that can 1) be used to predict future susceptibility of progression from colonization to infection for clinical studies and 2) be used for further basic studies to determine pathogenic mechanisms of action.

This reviewer is not a practicing bioinformaticist and thus I am not able to discretely judge all of the methods employed. I am, however, able to follow the basic premise of the study and have several questions or clarifications that, if answered/rectified, could further strengthen this manuscript:

-The abstract starts with "Klebsiella frequently..." The authors explain in the introduction that when they say *Klebsiella*, they are referring to *K. pneumoniae* complex. This, however, is not clear if just reading the abstract. I think it is important this is clearly stated so the abstract can stand alone. You would not want people to erroneously conclude this included *K. aerogenes* or *K. oxytoca*.

-I am a bit confused by the description of the isolates included as cases or controls. In figure 1B, you demonstrate how number of control and case rectal isolates increased similarly over the collection period. So in the end, how did you decide on 160 control rectal isolates compared to 85 case isolates? It is not clear if these were all from different patients; or were multiple different isolates ever used from

the same patient? Did you ever include non-infectious isolates from patients that when on to have a Klebsiella infection with a different strain?

-Right before introducing Suppl. Fig S7, you mention that many of the genes were highly associated with one another. You get into this a bit later with the plasmid discussion, but did you do any analysis to determine genetic distance between the progression-associated genes in the genome? You mention “collinear” in the discussion, but did you specifically assess this?

-When discussing the AMR genes, you refer to Kleborate throughout the paragraph. This tool should be referenced and/or explained here.

-Did the authors ever attempt to link the specific patient factors that they previously determined were associated with progression to infection with any of the Klebsiella genes identified in this study? I am not sure there is enough power to do this.

-The authors state that infection-associated genes are transmissible by plasmids, yet they only test conjugation of a single plasmid. Yes, the one plasmid tested is conjugative resulting in the transfer of infection-associated genes, but it is unclear how universal/generalizable this is.

-The use of a multiplex PCR to quickly identify six infection-associated genes is intriguing. Obviously, the use of such an assay to prospectively identify colonized patients that progress to Klebsiella infection would further corroborate this work.

-I believe “geographically-independent cohort” or “geographically-distinct cohort” should contain hyphens.

-Intro 2nd paragraph, line 4, should have a comma after “strains.”

In summary, this case-control comparative genomics study provides insight into Klebsiella genes associated with progression from gut colonization to distant infection. These data have implications for patient risk stratification, as well as basic understandings of pathogenic mechanisms of Klebsiella infection.

David A. Rosen, MD, PhD

Assistant Professor of Pediatric Infectious Diseases

Washington University School of Medicine

Reviewer #2 (Remarks to the Author):

I appreciate the chance to review this manuscript looking at genetic content between gut strains and later invasive strains of *Klebsiella* in patients from Michigan compared to a group from Maryland. I have a few comments for the authors.

First, it does seem that by only looking at concordant strains between gut and invasion, the study is set up to minimize variation and potentially miss genes which might be shared or transferred within the gut that could lead one strain to invade compared to another. Put another way, I wonder what would be found if one compared strains which only ever seemed to be colonizing yet never found as invasive? Or if one looked at the genetic content of all colonizing strains then looked at the invasive strains to see what gene transfer had occurred. I realize this is a completely different study, but I wonder about how the pre-selection of concordant strains may have biased the final results.

Second, I think a major potential confounder which is briefly addressed near the end of the discussion is that the genes in the invasive strains (such as antibiotic resistance genes, etc) may simply be markers for more resistant strains and that, in fact, these genes are not conferring a more invasive phenotype but are markers for sicker patients with more underlying comorbidities and more contact with healthcare - thus they're really a surrogate for the severity of illness in the patient. Although the general demographics are compared between groups, I see no control or accounting for measures of clinical severity or general clinical condition of the underlying patient. Work done by Bachman and others has shown before that often patient characteristics play a major role in determining who develops severe *Klebsiella* infections.

Third, and finally, I wish that, given the importance of plasmids and MGE apparently play in the findings, more than 10 strains had been selected for long read sequencing and hybrid genome assemblies. This is really key to fully defining the mechanisms at play which are being studied here. The relative lack of replicons in many strains (only found in 66.5%?) is somewhat concerning. Plasmids are notoriously hard to define with short read sequencing.

In all it is an interesting study, within the constraints listed, most of which are addressed, but I do think in light of the potential confounders the results have to be interpreted with caution and additional work could help further bolster the claims.

As a minor note, in the introduction, *K. quasivariicola* should still be labeled sp. nov. as the name has not yet been officially ratified although it is in common use.

Reviewer #3 (Remarks to the Author):

This tremendous paper by Vornhagen et al describes a nested case-control study to identify bacterial genetic factors associated with the transition of *Klebsiella* strains, belonging to the Kp species complex, from asymptomatic gut carriage to invasive infection. This remains a key question for *Klebsiella* control, as it will underpin much more rigorous risk assessments based on assessments of enteric flora. Although at its core the analysis is a fairly simple GWAS study, there are many additional aspects that add considerable novelty and power to the work, and the authors are to be congratulated in identifying, and successfully addressing, the many potential confounders that can limit the usefulness of this kind of comparative genomics approach. The main strengths of the study, as outlined in the introduction, are the use assembly of individual lineages both the colonisation (pre-infection) and infection stage from single patients, the use of matched controls and the statistical consideration of patient variables (including antibiotic exposure). The authors identify some key loci associated with infection risk, and there are many intriguing observations and questions - the fact that these genes tend to be plasmid borne, that some are associated with competitiveness (abundance) in the gut (but others are not), the role of AMR genes and the interplay between increased virulence and resistance. The work is really comprehensive, it seems every angle has been covered. The experimental data on transduction is reassuring, and the additional PCR data on a separate cohort is really important. As such, I found the paper fascinating, intriguing, thorough, well presented and easy to read, and - above all - convincing yet extremely well-balanced - there is no 'hard sell', all the limitations and caveats are laid bare (this was very refreshing!). The work also provides a fresh perspective in the importance of the well-known virulence factors (eg the siderophores) present within the hyper-virulent clones (ST23) which receive a lot of attention but in fact only account for a very small minority of disease cases. This of course also have implications for the generalisability of the 'virulence scores' generated using Kleborate. The emphasis on plasmids is also highly timely - given that we are now entering a new wave of 'long read' molecular data.

I only have minor queries and suggestions, and no serious concerns at all about the quality and importance of this paper.

Title - I think this undersells the paper - perhaps include some mention of the importance of AMR genes?

Line 52 – rephrase to ‘the high relative abundance of Klebsiella colonising the gut’.

Table 1 – I didn’t quite understand how there is an OR for age, as this is also (like sex) a matching variable.

The only potentially serious gap in the analysis I could identify was that the three species have been lumped together as one. I suspect this may actually be adding some noise to the Scoary analysis, as according to Figure 2 many of the associated genes tend to be less common in Kv and Kqp, including many of the AMR genes. It would therefore be interesting to speculate on whether these associations hold across all three species, or just Kp (similarly there may be other associated genes in Kv and Kqp that are not relevant for Kp) – I am not suggesting any more analysis, but maybe a line or two in the discussion where the limitations are addressed. Along the same lines- it may also turn out to be the case that case-control pairs also need to be matched in terms of bacterial species, not just patient variables.

I find it really curious that there is such a weak association between antibiotic exposure and associated resistance gene (Fig S.7) – perhaps the authors could expand a little on this – how surprising is it?

Response to reviewers

We thank the reviewers for their time and thoughtfulness in assessing our manuscript. We have carefully considered the reviewers' comments and concerns, and we believe that our manuscript is now significantly stronger. Our responses to the reviewers' comments and concerns are in blue italics and annotated throughout the marked-up manuscript document using track changes. Line numbers indicated through the reviewer response document refer to the clean manuscript document.

REVIEWER COMMENTS

Reviewer #1 (Remarks to the Author):

This study led by Jay Vornhagen of Michael Bachman's group at the University of Michigan examines Klebsiella isolates from colonized patients and performs a case-control comparative genomic analysis between those that went on to cause distant infection versus those that did not. Ultimately, the group identified 27 genes in the accessory genome that were associated with progression to infection. A subset of these genes were associated with gut abundance and a subset were not, suggesting they likely play a role in fitness at another point in pathogenesis. Interestingly, many of these genes were plasmid-borne.

This extremely-well written manuscript tackles an increasingly important issue in our patients. MDR Klebsiella strains are considered urgent or serious threats by the CDC, and Klebsiella is becoming one of our most troublesome nosocomial pathogens. Despite its importance, very little is understood about how this opportunist causes infection. Specifically, when a large fraction of society is colonized with Klebsiella (and an even larger fraction of hospitalized patients), how is it determined who goes on to suffer infection with Klebsiella? This group previously studied host factors that may make a patient more susceptible for progression to Klebsiella infection. Here, through WGS and a series of statistical models, they identify organism-specific factors that correlate with progression from colonization to infection. Overall, the work herein is well-described, well-controlled, transparent, rigorous, and important. While this study did not identify a "smoking gun" if you will, it did identify several Klebsiella factors that can 1) be used to predict future susceptibility of progression from colonization to infection for clinical studies and 2) be used for further basic studies to determine pathogenic mechanisms of action.

This reviewer is not a practicing bioinformaticist and thus I am not able to discretely judge all of the methods employed. I am, however, able to follow the basic premise of the study and have several questions or clarifications that, if answered/rectified, could further strengthen this manuscript:

-The abstract starts with "Klebsiella frequently..." The authors explain in the introduction that when they say Klebsiella, they are referring to K. pneumoniae complex. This, however, is not clear if just reading the abstract. I think it is important this is clearly stated so the abstract can stand alone. You would not want people to erroneously conclude this included K. aerogenes or K. oxytoca.

We agree that we should be clear that we studied the Klebsiella pneumoniae species complex. We have modified the abstract in accordance with this suggestion (ln. 27).

-I am a bit confused by the description of the isolates included as cases or controls. In figure 1B, you demonstrate how number of control and case rectal isolates increased similarly over the collection period. So in the end, how did you decide on 160 control rectal isolates compared to 85 case isolates? It is not clear if these were all from different patients; or were multiple different isolates ever used from the same patient? Did you ever include non-infectious isolates from patients that when on to have a Klebsiella infection with a different strain?

We apologize for this lack of clarity. Cases were selected based on careful chart review to ensure they met case definition of infection, and Figure 1B shows that they accumulated steadily over the course of enrolling colonized patients in our larger cohort study. All control isolates were from asymptotically colonized patients who never went on to have a Klebsiella infection. Our control selection was based on careful matching to cases by basic demographics (age, sex), and date of swab collection. So, the chosen controls accumulate at a similar rate to cases. We did not compare strains within a given patient.

-Right before introducing Suppl. Fig S7, you mention that many of the genes were highly associated with one another. You get into this a bit later with the plasmid discussion, but did you do any analysis to determine genetic

distance between the progression-associated genes in the genome? You mention “collinear” in the discussion, but did you specifically assess this?

We did not assess the genetic distance between infection-associated genes in a comprehensive way but agree that this is an interesting question. A limitation of our Illumina-based approach is that we cannot resolve the genetic distance between all infection-associated genes because the genome is broken into contigs. We did perform long-read sequencing on a subset of strains that contained several infection-associated genes, and this indicated that a given plasmid often contained multiple high-risk genes.

We replaced the term collinear, as we did not specifically test for collinearity. Rather, it is more appropriate to say that these genes are associated with one another (Ins. 471-473).

-When discussing the AMR genes, you refer to Kleborate throughout the paragraph. This tool should be referenced and/or explained here.

We have added a citation to all instances where we referenced Kleborate (Ins. 262, 268, 904).

-Did the authors ever attempt to link the specific patient factors that they previously determined were associated with progression to infection with any of the Klebsiella genes identified in this study? I am not sure there is enough power to do this.

We agree that this is an interesting question; however, as the reviewer indicates, we lack the power to perform this analysis. Our cohort study (doi 10.1128/mSphere.00132-21) was designed and powered for identifying important clinical variables, and we used those findings to inform how this study should adjust for potential confounders. We are excited by the opportunity to independently assess a relationship between our genes of interest and patient factors in future studies powered for this type of analysis.

-The authors state that infection-associated genes are transmissible by plasmids, yet they only test conjugation of a single plasmid. Yes, the one plasmid tested is conjugative resulting in the transfer of infection-associated genes, but it is unclear how universal/generalizable this is.

We have added more donor strains to assess the generalizability of this phenotype (Fig. 4D, Ins. 361-373). Indeed, we saw variable rates of conjugation based on donor and recipient strains but found that all plasmids tested could be transferred to at least one recipient. We have included additional discussion of the generalizability of our findings regarding HGT in the discussion (Ins. 474-482).

-The use of a multiplex PCR to quickly identify six infection-associated genes is intriguing. Obviously, the use of such an assay to prospectively identify colonized patients that progress to Klebsiella infection would further corroborate this work.

We agree. This is the goal of future work in the lab.

-I believe “geographically-independent cohort” or “geographically-distinct cohort” should contain hyphens.

This has been corrected (Ins. 36, 96, 375, 399, 404, 491, 905).

-Intro 2nd paragraph, line 4, should have a comma after “strains.”

This has been corrected (In. 64).

In summary, this case-control comparative genomics study provides insight into Klebsiella genes associated with progression from gut colonization to distant infection. These data have implications for patient risk stratification, as well as basic understandings of pathogenic mechanisms of Klebsiella infection.

David A. Rosen, MD, PhD
Assistant Professor of Pediatric Infectious Diseases

Reviewer #2 (Remarks to the Author):

I appreciate the chance to review this manuscript looking at genetic content between gut strains and later invasive strains of *Klebsiella* in patients from Michigan compared to a group from Maryland. I have a few comments for the authors.

First, it does seem that by only looking at concordant strains between gut and invasion, the study is set up to minimize variation and potentially miss genes which might be shared or transferred within the gut that could lead one strain to invade compared to another. Put another way, I wonder what would be found if one compared strains which only ever seemed to be colonizing yet never found as invasive? Or if one looked at the genetic content of all colonizing strains then looked at the invasive strains to see what gene transfer had occurred. I realize this is a completely different study, but I wonder about how the pre-selection of concordant strains may have biased the final results.

*We agree that this is an interesting question and would be an interesting study to perform. As the reviewer correctly indicates, we chose to focus on concordant cases. This was for two reasons. One, we sought to identify genes associated with the transition from colonization to infection. We feel this is a fundamental gap in knowledge of *Klebsiella* pathogenesis that we can begin to address. It also moves us towards our long-term goal of being able to predict infections based on characterization of colonizing isolates and enabling infection prevention interventions. Second, we wanted to minimize variation and confounding. We chose to focus on concordant cases to be sure that our cases were infected from the gut reservoir, rather than an external source, and extensively controlled for corresponding metadata. Infecting strains that are discordant from those found on the patient's rectal swab may have been introduced into the site of infection from the hospital environment and not be gut colonizers. Although our approach did not specifically evaluate transfer of genes between colonizing strains, it also does not exclude the possibility that this occurred. We used the rectal swab that was the closest in time to the clinical isolate, so it is possible that genetic exchange occurred prior to this date. We agree that our approach limits the breadth of the pangenome for case isolates, and our matching criteria limits the pangenome for control isolates, and therein narrows the ability to comprehensively detect HGT events. We have included further discussion of this limitation of our study (lns. 474-484). We aspire to undertake a more comprehensive study to understand the role gene transfer within the gut and how that influences infection risk in the future.*

Second, I think a major potential confounder which is briefly addressed near the end of the discussion is that the genes in the invasive strains (such as antibiotic resistance genes, etc) may simply be markers for more resistant strains and that, in fact, these genes are not conferring a more invasive phenotype but are markers for sicker patients with more underlying comorbidities and more contact with healthcare - thus they're really a surrogate for the severity of illness in the patient. Although the general demographics are compared between groups, I see no control or accounting for measures of clinical severity or general clinical condition of the underlying patient. Work done by Bachman and others has shown before that often patient characteristics play a major role in determining who develops severe *Klebsiella* infections.

*We agree that patient characteristics are critical determinants of infection risk and apologize that our approach for controlling for comorbidities was not clear. Since this is a necessarily small study due to the cost of sequencing, it has insufficient power to adequately model the myriad clinical variables that can impact infection risk in a cohesive way and would be subject to inaccuracies. Therefore, we used our prior data and a larger cohort to derive what clinical variables were important to consider as confounders, and then we used these variables in the smaller study to generate scores alongside matching and a weighted regression to control for these variables. This approach controls for independent clinical confounders while still preserving statistical power for our central questions regarding the risk of infection in relation to *Klebsiella* genetics. Thus, starting with a cohort of nearly 2,000 colonized patients, we identified overall comorbid disease burden, as measured by a weighted Elixhauser score, specific medications including antibiotics, low albumin, and others (doi 10.1128/mSphere.00132-21) as important independent clinical risk factors for infection. While many of these variables are linked to increased healthcare utilization, we did not find healthcare exposure itself to be an*

independent risk factor in this analysis. We then used this information from the larger cohort to adjust for those clinical differences in this nested case-control study. We did separately evaluate weighted Elixhauser score as a measure of severity of illness and found a trend but no statistical difference between cases and controls (Table 1). Regardless, for each gene identified on unadjusted analysis, we followed this up with a weighted conditional regression, with weights derived from comorbidities alongside the other important clinical predictors and found that antibiotic resistance genes were still associated with infection after this adjustment. We have clarified that comorbidities were adjusted for in the results section (ln. 216).

Third, and finally, I wish that, given the importance of plasmids and MGE apparently play in the findings, more than 10 strains had been selected for long read sequencing and hybrid genome assemblies. This is really key to fully defining the mechanisms at play which are being studied here. The relative lack of replicons in many strains (only found in 66.5%?) is somewhat concerning. Plasmids are notoriously hard to define with short read sequencing.

We agree that long read sequencing has become the gold standard to map genes to plasmids versus the chromosome. Our goal in long-read sequencing of a subset of strains was to provide initial characterization of the genetic context of these infection-associated genes. We feel that this was successful in demonstrating that most infection-associated genes can be found on plasmids, and that there was significant diversity of the gene content of these plasmids, even across a small number of strains. Inclusion of additional hybrid assemblies would extend our findings but would not affect our main finding of specific genes associated with infection. However, comprehensive long-read sequencing was not originally budgeted for and would cost well over \$10,000. Unfortunately, we currently do not have the budget for this experiment. It is certainly true that we could be missing certain plasmids due to the limitations of the PlasmidFinder tool, although other studies have noted that some Klebsiella strains lack plasmids (PMID 29723841). We have added further discussion of this limitations of PlasmidFinder and the absence of comprehensive long-read sequencing in our study (lns. 474-482).

In all it is an interesting study, within the constraints listed, most of which are addressed, but I do think in light of the potential confounders the results have to be interpreted with caution and additional work could help further bolster the claims.

As a minor note, in the introduction, *K. quasivariicola* should still be labeled sp. nov. as the name has not yet been officially ratified although it is in common use.

This has been corrected (ln. 44).

Reviewer #3 (Remarks to the Author):

This tremendous paper by Vornhagen et al describes a nested case-control study to identify bacterial genetic factors associated with the transition of *Klebsiella* strains, belonging to the Kp species complex, from asymptomatic gut carriage to invasive infection. This remains a key question for *Klebsiella* control, as it will underpin much more rigorous risk assessments based on assessments of enteric flora. Although at its core the analysis is a fairly simply GWAS study, there are many additional aspects that add considerable novelty and power to the work, and the authors are to be congratulated in identifying, and successfully addressing, the many potential confounders that can limit the usefulness of this kind of comparative genomics approach. The main strengths of the study, as outlined in the introduction, are the use assembly of individual lineages both the colonisation (pre-infection) and infection stage from single patients, the use of matched controls and the statistical consideration of patient variables (including antibiotic exposure). The authors identify some key loci associated with infection risk, and there are many intriguing observations and questions - the fact that these genes tend to be plasmid borne, that some are associated with competitiveness (abundance) in the gut (but others are not), the role of AMR genes and the interplay between increased virulence and resistance. The work is really comprehensive, it seems every angle has been covered. The experimental data on transduction is reassuring, and the additional PCR data on a separate cohort is really important. As such, I found the paper fascinating, intriguing, thorough, well presented and easy to read, and - above all - convincing yet extremely well-balanced - there is no 'hard sell', all the limitations and caveats are laid bare (this was very refreshing!). The work also provides a fresh perspective in the importance of the well-known virulence factors (eg the

siderophores) present within the hyper-virulent clones (ST23) which receive a lot of attention but in fact only account for a very small minority of disease cases. This of course also have implications for the generalisability of the 'virulence scores' generated using Kleborate. The emphasis on plasmids is also highly timely – given that we are now entering a new wave of 'long read' molecular data. I only have minor queries and suggestions, and no serious concerns at all about the quality and importance of this paper.

Title – I think this undersells the paper – perhaps include some mention of the importance of AMR genes?

We thank the reviewer for their enthusiasm about the significance of our findings. We feel that the association of AMR genes with infection is nuanced and would not be easily expressed in a title. To emphasize our control for potential confounders, we propose:

“Combined comparative genomics and clinical modelling reveals plasmid-encoded genes are independently associated with Klebsiella infection”

Line 52 – rephrase to ‘the high relative abundance of Klebsiella colonising the gut’.

This has been corrected (Ins. 52-53).

Table 1 – I didn't quite understand how there is an OR for age, as this is also (like sex) a matching variable.

ORs for variables used to match cases and controls have been removed (Table 1).

The only potentially serious gap in the analysis I could identify was that the three species have been lumped together as one. I suspect this may actually be adding some noise to the Scoary analysis, as according to Figure 2 many of the associated genes tend to be less common in Kv and Kqp, including many of the AMR genes. It would therefore be interesting to speculate on whether these associations hold across all three species, or just Kp (similarly there may be other associated genes in Kv and Kqp that are not relevant for Kp) – I am not suggesting any more analysis, but maybe a line or two in the discussion where the limitations are addressed. Along the same lines- it may also turn out to be the case that case-control pairs also need to be matched in terms of bacterial species, not just patient variables.

We also carefully weighed the benefits and drawbacks of analyzing these three species as a group. We ultimately decided that since these species caused similar disease and exchanged genes in the accessory genome, we should consider them together. Although not reported in the manuscript, we attempted the Scoary analysis stratified by species. Limiting this analysis to Kp did reveal some novel genes associated with infection; however, many of these were low frequency (<10%) in our dataset. Conversely, many of the higher-frequency genes had similar point estimates for association with infection following exclusion of Kv and Kqp (Reviewer Table 1). Therefore, these findings are likely to apply to K. pneumoniae, and our validation at a second site (Table 4) indicates that at least a subset are reproducibly associated with infection at the level of the species complex. However, the study was underpowered to identify strong associations for Kv and Kqp. We have added the low Kv and Kqp prevalence as a limitation to this study (In. 484-486).

I find it really curious that there is such a weak association between antibiotic exposure and associated resistance gene (Fig S.7) – perhaps the authors could expand a little on this – how surprising is it?

Our original assumption was that the association between antibiotic resistance genes and infection would be lost following adjustment for high-risk antibiotic exposure. Some genes, such as qacED1 did follow this assumption, (Table 2); however, we were surprised that this was not the case for all antibiotic resistance genes. This suggests a mechanism other than selective pressure from antibiotic exposure that resulted in these genes being identified as significant predictors of infection. One possible interpretation of this result, which we are excited by, is that it indicates several antibiotic resistance genes were present on plasmids containing other infection-associated genes that are functionally increasing the risk of infection. Therefore, detection of these antibiotic resistance genes is more likely a secondary marker of other infection-associated genes and/or plasmids. It is worth noting that antibiotic exposure rates are low (<10% for most classes) in our study population (PMID 34160237). As such, it is likely the case that many of the patients colonized by a Klebsiella strain encoding antibiotic resistance

genes are not exposed to a cognate antibiotic, and these strains are acquiring these genes at some other time in their evolutionary history. We have added additional language to the discussion to highlight this point (Ins. 438-441).

Reviewer Table 1. Unit frequency and association with case status after exclusion of *K. variicola* and *K. quasipneumoniae*

Locus	Non-unique Gene name	Annotation	Kp frequency	Kp count	Kv frequency	Kv count	Kqp frequency	Kqp count	Total frequency	Total count	Odds ratio: All isolates	Odds ratio: Kp only
aac(6)-Ib-cr5		fluoroquinolone-acetylating aminoglycoside 6'-N-acetyltransferase AAC(6)-Ib-cr5	0.087	16	0.000	0	0.000	0	0.065	16	30.68	30.82
bla_{OXA-1}		oxacillin-hydrolyzing class D β-lactamase OXA-1	0.076	14	0.000	0	0.000	0	0.057	14	25.54	25.77
bla_{CTX-M-15}	bla_2	class A extended-spectrum β-lactamase CTX-M-15	0.087	16	0.000	0	0.000	0	0.065	16	13.94	14.00
group_13461		IS1380 family transposase ISEcp1	0.087	16	0.000	0	0.000	0	0.065	16	13.94	14.00
group_13460		hypothetical protein	0.082	15	0.000	0	0.000	0	0.061	15	12.69	12.77
group_13458		hypothetical protein	0.076	14	0.000	0	0.000	0	0.057	14	11.46	11.59
bla_{TEM-1}		class A broad-spectrum β-lactamase TEM-1	0.114	21	0.023	1	0.111	2	0.098	24	6.28	8.82
group_7606		hypothetical protein	0.071	13	0.047	2	0.056	1	0.065	16	6.26	4.06
tnpR		Transposon Tn3 resolvase	0.103	19	0.000	0	0.056	1	0.082	20	5.97	7.50
group_13437		hypothetical protein	0.103	19	0.023	1	0.000	0	0.082	20	5.97	5.45
group_13467		IS6 family transposase IS6100	0.109	20	0.093	4	0.111	2	0.106	26	5.75	8.15
group_5875--tehA_2--group_10494		HTH-type transcriptional regulatory protein GabR--Tellurite resistance protein TehA--hypothetical protein	0.082	15	0.000	0	0.000	0	0.061	15	5.65	5.13
qacEΔ1		quaternary ammonium compound efflux SMR transporter QacE delta 1	0.082	15	0.047	2	0.056	1	0.073	18	5.46	3.75
group_9575	xerC_6	Tyrosine recombinase XerC	0.141	26	0.070	3	0.111	2	0.127	31	5.39	5.69
group_5455-sul2		IS91 family transposase - sulfonamide-resistant dihydropteroate synthase Sul2	0.071	13	0.023	1	0.056	1	0.061	15	5.07	6.17
sul1		sulfonamide-resistant dihydropteroate synthase Sul1	0.082	15	0.023	1	0.056	1	0.069	17	4.97	3.63
group_13965		hypothetical protein	0.103	19	0.000	0	0.000	0	0.078	19	4.11	4.11
aqpZ_2		Aquaporin Z	0.179	33	0.047	2	0.056	1	0.147	36	3.96	2.66
group_13443		hypothetical protein	0.168	31	0.000	0	0.000	0	0.127	31	3.81	3.17
group_9537		hypothetical protein	0.174	32	0.000	0	0.056	1	0.135	33	3.71	2.89
dgcE		hypothetical protein	0.261	48	0.047	2	0.056	1	0.208	51	3.37	2.78
arsD_1		Arsenical resistance operon trans-acting repressor ArsD	0.234	43	0.070	3	0.222	4	0.204	50	3.19	2.96
group_2409		ISL3 family transposase ISEc38	0.190	35	0.070	3	0.056	1	0.159	39	3.07	2.64
pdeC_2		putative cyclic di-GMP phosphodiesterase PdeC	0.272	50	0.093	4	0.222	4	0.237	58	2.88	2.79
group_9528		hypothetical protein	0.277	51	0.047	2	0.167	3	0.229	56	2.87	2.64
group_622		hypothetical protein	0.413	76	0.116	5	0.167	3	0.343	84	2.71	2.16
group_7231		hypothetical protein	0.234	43	0.047	2	0.167	3	0.196	48	2.69	2.61
group_9530		hypothetical protein	0.364	67	0.186	8	0.111	2	0.314	77	2.64	2.83
ltrA		hypothetical protein	0.348	64	0.140	6	0.111	2	0.294	72	2.57	2.39
group_774		hypothetical protein	0.353	65	0.279	12	0.111	2	0.322	79	2.55	3.10
group_4187	kIcA_2	Antirestriction protein KlcA	0.353	65	0.093	4	0.111	2	0.290	71	2.54	2.80
umuC_2		Protein UmuC	0.641	118	0.209	9	0.222	4	0.535	131	2.52	2.64
umuD_2		Protein UmuD	0.538	99	0.186	8	0.167	3	0.449	110	2.42	2.44
group_2513		hypothetical protein	0.190	35	0.163	7	0.000	0	0.171	42	0.26	0.16

REVIEWERS' COMMENTS

Reviewer #2 (Remarks to the Author):

I appreciate the authors attention to the review comments and look forward to seeing their follow-up studies based upon this work.

RESPONSE TO REVIEWERS' COMMENTS

Reviewer #2 (Remarks to the Author):

I appreciate the authors attention to the review comments and look forward to seeing their follow-up studies based upon this work.

We thank the reviewer for their thorough and constructive feedback that helped us to improve this article.